# The proneural wave in the *Drosophila* optic lobe is driven by an excitable reaction-diffusion mechanism

**David J Jörg[1,2], Elizabeth E Caygill[2,3], Anna E Hakes[2,3], Esteban G Contreras[2], Andrea H Brand[2], Benjamin D Simons[1,2,4]***

[1]Cavendish Laboratory, Department of Physics, University of Cambridge, Cambridge, United Kingdom; [2]The Wellcome Trust/Cancer Research UK Gurdon Institute, University of Cambridge, Cambridge, United Kingdom; [3]Department of Physiology, Development and Neuroscience, University of Cambridge, Cambridge, United Kingdom; [4]The Wellcome Trust/Medical Research Council Stem Cell Institute, University of Cambridge, Cambridge, United Kingdom

**Abstract** In living organisms, self-organised waves of signalling activity propagate spatiotemporal information within tissues. During the development of the largest component of the visual processing centre of the *Drosophila* brain, a travelling wave of proneural gene expression initiates neurogenesis in the larval optic lobe primordium and drives the sequential transition of neuroepithelial cells into neuroblasts. Here, we propose that this 'proneural wave' is driven by an excitable reaction-diffusion system involving epidermal growth factor receptor (EGFR) signalling interacting with the proneural gene *l'sc*. Within this framework, a propagating transition zone emerges from molecular feedback and diffusion. Ectopic activation of EGFR signalling in clones within the neuroepithelium demonstrates that a transition wave can be excited anywhere in the tissue by inducing signalling activity, consistent with a key prediction of the model. Our model illuminates the physical and molecular underpinnings of proneural wave progression and suggests a generic mechanism for regulating the sequential differentiation of tissues.
DOI: https://doi.org/10.7554/eLife.40919.001

**\*For correspondence:**
bds10@cam.ac.uk

**Competing interests:** The authors declare that no competing interests exist.

## Introduction

The development of multicellular organisms relies on a multitude of transient coordination processes that provide the spatiotemporal cues for cell fate decision-making and thereby ensure that tissues are specified with the correct size, pattern and composition (*Perrimon et al., 2012*; *Oates et al., 2012*; *Sato et al., 2013*). In one strategy, large-scale patterning is engineered by self-organised concentration waves of biomolecular fate determinants that travel across tissues through intercellular exchange and the regulation of gene expression. Such travelling waves, which are viable carriers of spatiotemporal information, are a ubiquitous feature of developmental pattern formation, where they arise through different underlying mechanisms, from coordinated intracellular oscillations (*Oates et al., 2012*; *Jörg et al., 2015*; *Hubaud et al., 2017*; *Verd et al., 2018*) to self-organised reaction-diffusion processes (*Lubensky et al., 2011*; *Formosa-Jordan et al., 2012*; *Fried et al., 2016*; *Gavish et al., 2016*; *Corson et al., 2017*).

During the development of the fruitfly *Drosophila melanogaster*, a propagating wave of gene expression orchestrates the patterning of the largest component of the visual processing centre: neuroepithelial cells in the optic lobe of the larval brain divide symmetrically, expanding the progenitor pool, and then undergo a sequential transition into asymmetrically dividing neuroblasts, which generate the neurons of the medulla (*Figure 1a,b*) (*Egger et al., 2007*; *Yasugi et al., 2008*;

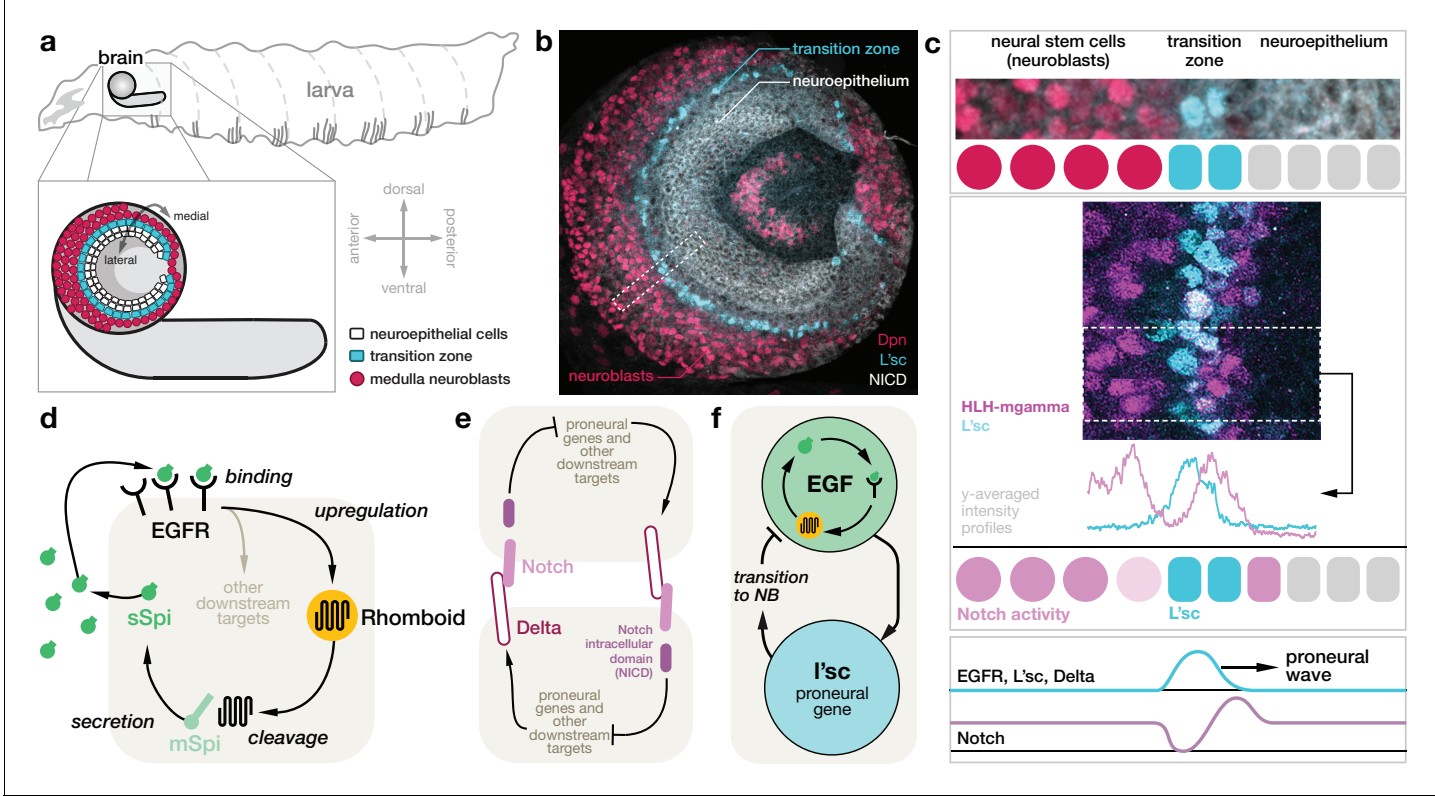

**Figure 1.** Molecular basis for the proneural wave in the *Drosophila* optic lobe. (**a**) Schematic depiction of the *Drosophila* larva at the late 3rd instar stage when the proneural wave is transforming the neuroepithelium into medulla neuroblasts. (**b**) Optic lobe in a lateral view showing the neuroepithelium (labelled with Notch intracellular domain (NICD), white), the transition zone (L'sc, cyan) and the neuroblasts (Dpn, red). (**c**) L'sc expression and Notch signalling activity around the transition zone. Top: Magnification of the region outlined in (b), showing neuroblasts (Dpn, red), L'sc expression (cyan) and the neuroepithelium (NICD, white). Middle: Confocal image showing that Notch signalling activity (HLH-mgamma, purple) increases just before the transition zone (marked by L'sc, cyan), drops during the transition and then increases again in neuroblasts. Bottom: The proneural wave, characterised by expression of L'sc as well as EGF receptor (EGFR) and Notch signalling activity, sequentially converts the neuroepithelium into neuroblasts. (**d**) EGFR signalling in the transition zone activates expression of the transmembrane protein Rhomboid, which in turn cleaves the membrane-tethered form of the EGFR ligand Spitz (mSpi) to generate its active secreted form (sSpi). (The shaded region depicts an individual cell in the neuroepithelium.) sSpi can bind to the EGFR on the same cell and neighbouring cells. (**e**) Delta-Notch signalling is a contact-dependent signalling pathway active in both the neuroepithelium and the neuroblasts. The Delta ligand binds to Notch receptors on adjacent cells upon which their intracellular domain (NICD) is cleaved. The NICD regulates target genes, which, in turn, affects expression of Delta. (**f**) Active EGFR signalling promotes the expression of L'sc within the same cell, which is sufficient for the neuroepithelium to neuroblast transition and which in turn downregulates EGFR signalling.

DOI: https://doi.org/10.7554/eLife.40919.002

*Egger et al., 2010*; *Yasugi et al., 2010*). The transition from neuroepithelial cells to neuroblasts occurs at a 'transition zone' that sweeps from one side of the optic lobe to the other and is marked by the expression of the proneural gene lethal of scute (*l'sc*) (*Yasugi et al., 2008*). This 'proneural wave' is controlled by the coordinated action of different signalling pathways: epidermal growth factor receptor (EGFR) signalling and Delta-Notch signalling (*Figure 1c–f*) (*Yasugi et al., 2010*). The sequential nature of the transition is crucial to generate populations of cells of different developmental ages that give rise to a diverse array of terminally differentiated medulla neurons (*Li et al., 2013*; *Sato et al., 2013*; *Suzuki et al., 2013*; *Erclik et al., 2017*). The transition zone exhibits localised EGFR signalling as well as expression of *l'sc* (*Figure 1b,c*). Absence of EGFR signalling leads to loss of the differentiation wave, indicating that EGFR signalling is a key component for proneural wave progression (*Yasugi et al., 2010*). The neuroepithelium exhibits low levels of Notch signalling activity (*Egger et al., 2011*; *Weng et al., 2012*). However, Notch activity peaks directly before the transition from neuroepithelial cell to neuroblast, drops during the transition and then is restored upon neuroblast transformation (*Figure 1c*) (*Contreras et al., 2018*). In addition, coordinating roles are played

by the JAK/STAT and Fat-Hippo pathways, which are broadly expressed in the neuroepithelium and prevent premature and ectopic transition of the neuroepithelium (*Yasugi et al., 2008*; *Yasugi et al., 2010*; *Wang et al., 2011a*; *Reddy et al., 2010*; *Kawamori et al., 2011*; *Weng et al., 2012*; *Tanaka et al., 2018*).

The question of how the specific functional feedbacks of EGFR signalling and proneural gene expression generate a localised propagating transition zone requires a mechanistic explanation of wave progression based on molecular feedbacks and signalling cascades. Such a description should explain (i) the dynamic nature of the wave, (ii) the emergence of a localised transition zone with spatially confined expression of the proneural gene *l'sc* and (iii) the specific profiles of gene expression and signalling activity around the transition zone. Moreover, the nature and function of the interaction of these components with Delta-Notch signalling, more commonly associated with lateral inhibition of neighbouring cells, is poorly understood, see Appendix 3. While a recent effort of a phenomenological description of the proneural wave (*Sato et al., 2016*) has started to model the coarse-grained aspects of proneural wave progression, the emergence of some major characteristics of the wave (such as spatially confined proneural gene expression in a localised transition zone) has not been addressed. Here we propose a model of signalling activity and proneural gene expression that describes the emergence of the proneural wave. Within this framework, the neuroepithelium behaves as an excitable medium in which changes in gene expression at the tissue boundary initiate a spontaneous wave of signalling activity that effects the transition from neuroepithelium to neuroblasts.

## Results

### Travelling front model of EGFR signalling activity

To develop the model, we first considered interactions between L'sc expression and associated signalling pathways within the transition zone. Previously, it was proposed that sequential induction of EGFR signalling is responsible for the progression of the proneural wave (*Yasugi et al., 2008*). EGFR signalling activates the expression of L'sc (*Yasugi et al., 2010*), which is sufficient to drive the neuroepithelium to neuroblast (NE to NB) transition (*Yasugi et al., 2008*; *Contreras et al., 2018*). The EGFR is activated by binding the secreted form of its ligand, Spitz. Secreted Spitz is generated by cleavage of a membrane-bound precursor by the transmembrane protease, Rhomboid (*Klämbt, 2000*). EGFR, Rhomboid and secreted Spitz together form an autocrine positive feedback loop (*Figure 1d*) (*Wiley et al., 2003*; *Sato et al., 2013*). In a first step, we noted that the dynamics of EGFR signalling alone has features that are sufficient to enable such a sequential induction and produce a travelling front of EGFR signalling activity; a feature notably absent in recent attempts to model the proneural wave, which also require further components to stabilise the propagating EGFR signalling front (*Sato et al., 2016*). In a minimal model based on the EGFR/Rhomboid/Spitz positive feedback loop, EGFR signalling activity is represented by a single component 'E' (e.g., the local cellular concentration of the active form of Spitz) that is diffusible between cells and involves the aforementioned positive feedback (*Figure 1d* and *Figure 2a*; Appendix 1),

$$\frac{\partial \phi^{\mathrm{E}}}{\partial t} = \eta \nabla^2 \phi^{\mathrm{E}} + \mu h(\phi^{\mathrm{E}}) - k\phi^{\mathrm{E}} \, . \tag{1}$$

Here $\phi^{\mathrm{E}}$ denotes the local strength of E activity, $\eta$ is the effective diffusion constant, $\mu$ is the gain rate in signalling activity due to positive feedback, $k$ is the decay rate, and $h(\phi) = \phi^n/(1 + \phi^n)$ is a Hill function parameterising the nonlinear positive feedback. Generically, reaction-diffusion systems involving diffusion and self-activation are known to support travelling bistable fronts that leave behind an elevated signalling state (*Figure 2a*; *Video 1*; Appendix 1) (*Muratov and Shvartsman, 2004*; *Keener and Sneyd, 2009*; *Graham et al., 2010*; *Tayar et al., 2015*). EGFR signalling is a natural candidate for being a key driver of the proneural wave and experimental evidence has shown that EGFR signalling is both necessary and sufficient for wave progression (*Yasugi et al., 2010*).

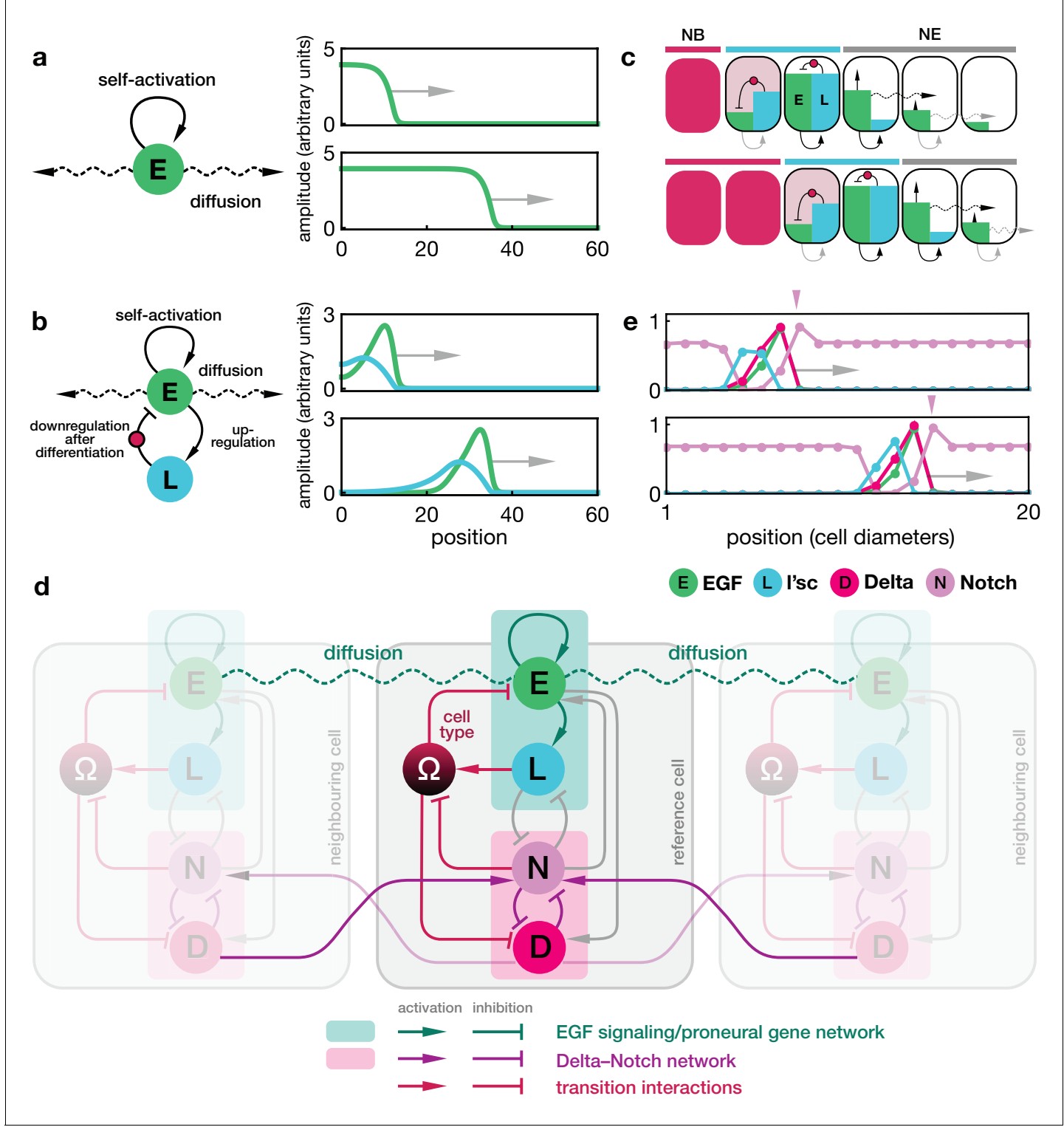

**Figure 2.** Dynamics of the proneural wave as an excitable reaction-diffusion system. (**a**) Minimal model of EGFR signalling. The dynamics of the EGFR/ Rhomboid/Spitz feedback loop is condensed in a single component 'E' (green), which is diffusible between cells and able to self-activate (for details, see Appendix 1). This single component represents a proxy for the activity of the feedback loop shown in *Figure 1c*, for example the local concentration of the active form of Spitz. The corresponding reaction-diffusion system, *Equation 1*, can give rise to a propagating front that leaves behind an elevated signalling state. Plots show the numerical solution of *Equation 1* in a one-dimensional representation of tissue (for simplicity) at two time points with initially elevated levels of E at the left-hand side of the domain. Specifying position in units of the diffusion length $\sqrt{\eta/k}$ and time in

*Figure 2 continued on next page*

*Figure 2 continued*

units of the decay time $k^{-1}$, the remaining chosen parameters are $\mu = 4$ and $n = 3$. (**b**) Model of EGFR signalling interacting with the proneural gene *l'sc*. EGFR signalling activates L'sc expression (component 'L', blue), which effectively inhibits EGFR signalling by driving the NE to NB transition (for details, see Appendix 2). The corresponding reaction-diffusion system, ***Equation 2***, can give rise to a propagating localised pulse of signalling activity and proneural gene expression corresponding to the transition zone. Parameters for E are the same as in panel a; parameters for L are $\mu_L = 0.4$, $k_L = 0.2$. (**c**) Schematic depiction of the mechanism giving rise to a localised transition zone, shown in panel b. Diffusion of signalling components (E, green) into the neuroepithelium leads to activation of the positive feedback loop, which locally excites signalling and proneural gene expression (L, blue) (Materials and methods). The excitation ceases as downregulation of signalling occurs, a consequence of the transition triggered by L'sc expression. (**d**) Regulatory network of the refined model including Delta-Notch signalling (D and N) and a local variable $\Omega$ indicating the cell state ($\Omega = 0$ indicates neuroepithelial cells and $\Omega = 1$ indicates neuroblasts; for details, see Appendix 3). Each shaded cell indicates one lattice site corresponding to one cell of the tissue. (**e**) Simulation of the integrated model of EGFR signalling, L'sc expression, Delta-Notch signalling and the NE to NB transition in a one-dimensional array of cells. The emerging spatial signalling and gene expression profile is characterised by a pulse of EGFR signalling, L'sc and Delta, and a drop in Notch signalling activity within the transition zone. The drop in Notch is preceded by a pulse of Notch signalling activity (pink arrowheads), which is due to a local lateral inhibition effect mediated by Delta-Notch signalling. Parameters are given in ***Appendix 3—table 1*** except for $\eta = 0.03$.

DOI: https://doi.org/10.7554/eLife.40919.003

## Travelling pulse model of EGFR signalling and proneural gene expression

However, notably, the EGFR/Rhomboid/Spitz feedback loop does not remain active in the wake of the travelling wave, but remains spatially confined as Rhomboid is expressed only transiently in the travelling transition zone (***Yasugi et al., 2010***; ***Sato et al., 2013***). Therefore, in a second step, we considered the influence of the proneural gene *l'sc*, represented by a second component 'L' in our model. Elevated EGFR signalling activates L'sc expression (***Yasugi et al., 2010***), which is sufficient to drive the NE to NB transition (***Yasugi et al., 2008***; ***Contreras et al., 2018***). In this minimal model, L'sc downregulates EGFR signalling as a consequence of the transition, leading to an indirect negative feedback (***Figure 1f***). The corresponding reaction-diffusion system for the local strength of E and L activity, $\phi^E$ and $\phi^L$, is given by

$$
\begin{aligned}
\frac{\partial \phi^E}{\partial t} &= \eta \nabla^2 \phi^E + \mu_E h(\phi^E)[1 - h(\phi^L)] - k_E \phi^E , \\
\frac{\partial \phi^L}{\partial t} &= \mu_L h(\phi^E) - k_L \phi^L ,
\end{aligned}
\tag{2}
$$

where $\mu_i$ (with $i = E, L$) indicate production rates and $k_i$ denote decay rates. Simulations of ***Equation 2*** demonstrate that this type of feedback is sufficient to describe a travelling localised pulse of signalling activity and L'sc expression through the tissue (***Figure 2b,c***; ***Video 2***; Appendix 2). Notably, the dynamics of L'sc (L) alter the bistable signalling behaviour of EGFR signalling (E) into an excitable one: once sufficiently perturbed by diffusion from an adjacent cell with an elevated signalling state, the intracellular reaction dynamics produces a transient expression pulse that downregulates itself as a result of the NE to NB cell fate transition (Appendix 2).

## Integrated model of the proneural wave to include EGFR-L'sc-Notch interactions

We next aimed to develop a more refined model that could be challenged by experiment and compared with previously published data. Such a model necessarily includes Delta-Notch signalling, which has been shown to influence how long cells remain in the L'sc expressing state (***Yasugi et al., 2010***; ***Wang et al., 2011b***; ***Weng et al., 2012***). As a mediator of lateral inhibition, Delta-Notch signalling is often associated with the emergence of 'salt-and-pepper'-like patterns of cell fate (***Bray, 2006***; ***Shaya and Sprinzak, 2011***). However, this pattern is not

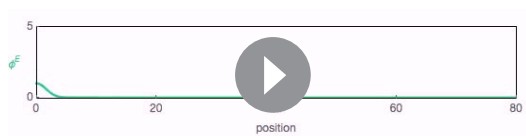

**Video 1.** Travelling EGFR signalling front. The movie shows the simulation of a one-dimensional version of the EGFR signalling model ***Equation 1*** corresponding to the snapshots shown in ***Figure 2a***. All simulation parameters as in ***Figure 2a***.

DOI: https://doi.org/10.7554/eLife.40919.004

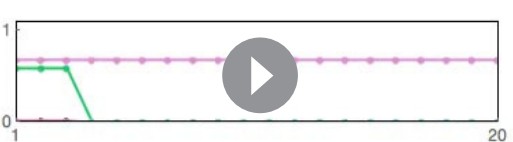

**Video 2.** Travelling EGFR signalling pulse and proneural gene expression. The movie shows the simulation of a one-dimensional version of the model of EGFR signalling interacting with the proneural gene *l'sc*, *Equation 2*, corresponding to the snapshots shown in main text *Figure 2b*. All simulation parameters as in *Figure 2b*.
DOI: https://doi.org/10.7554/eLife.40919.005

seen during proneural wave progression and the reasons for this are not clear (*Egger et al., 2010*; *Pérez-Gómez et al., 2013*; *Sato et al., 2016*). To address this question, we extended our minimal model to include canonical Delta-Notch interactions (*Figure 2d*) (*Collier et al., 1996*; *Bray, 2006*; *Simakov and Pismen, 2013*): (i) trans-activation of Notch by Delta, (ii) downregulation of Delta by Notch within the same cell and (iii) cis-inhibition (downregulation of Notch by Delta in the same cell). The model incorporates interactions between the Delta-Notch signalling pathway, EGFR signalling and L'sc expression, namely, upregulation of Delta through EGFR signalling (*Yasugi et al., 2010*),

upregulation of EGFR signalling through Notch signalling (*Yasugi et al., 2010*), downregulation of L'sc through Notch signalling (*Reddy et al., 2010*) and downregulation of Notch expression through L'sc (*Egger et al., 2010*). Despite these complex interactions, the functional 'module' comprising EGFR signalling and L'sc expression still remains the driver of the wave (green box in *Figure 2d*), while Delta-Notch signalling acts to provide further timing cues for the transition and to prevent premature differentiation (pink box in *Figure 2d*) (*Egger et al., 2010*; *Reddy et al., 2010*; *Yasugi et al., 2010*). The integrated model also includes an explicit representation of the cell state dynamics during the transition between neuroepithelial cell to neuroblast. The NE to NB transition is promoted by L'sc (*Yasugi et al., 2008*) and downregulation of Notch in the presence of EGFR signalling (*Yasugi et al., 2010*; *Weng et al., 2012*). The mathematical details of the refined model are given in Appendix 3.

## Congruence with experimental data

In addition to the emergence of a propagating transition zone, the integrated model also yielded predictions on the spatial profiles of signalling activity and gene expression (*Figure 2e*; Figure 4a; *Video 3*; *Video 4*; Appendix 4), which were in striking agreement with features observed in prior experiments. First, EGFR signalling, as well as L'sc and Delta expression, was found to be elevated only in the transition zone (*Figure 1c*) (*Egger et al., 2010*; *Yasugi et al., 2010*). Second, a peak of Notch activity is observed slightly in advance of the transition zone (pink arrowheads in *Figure 2e*) followed by a sharp drop in Notch activity (*Figure 1c*) (*Egger et al., 2011*; *Orihara-Ono et al., 2011*; *Weng et al., 2012*; *Contreras et al., 2018*). According to the model, lateral inhibition promotes high-Delta/low-Notch and low-Delta/high-Notch states in adjacent cells, and leads to a travelling 'laterally inhibited' cell state as the wave progresses. By contrast, the drop in Notch levels in transitioning cells arises due to cis-inhibition in our model as a consequence of Delta binding to Notch within the same cell, as has been shown experimentally (*Reddy et al., 2010*; *Weng et al., 2012*; *Contreras et al., 2018*).

To challenge the model further, we checked whether the documented effects of misregulating EGFR signalling, Notch signalling or L'sc expression (*Yasugi et al., 2010*) could be reproduced.

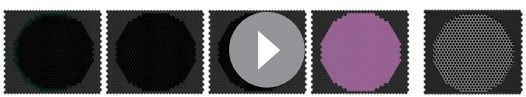

**Video 3.** Travelling proneural wave in the integrated model on a 1D array. The movie shows a simulation of the proneural wave model *Equations 17–19* simulated on a one-dimensional array of cells. All simulation parameters as in *Figure 2e*.
DOI: https://doi.org/10.7554/eLife.40919.006

**Video 4.** Travelling proneural wave in the integrated model on a 2D hexagonal lattice. The movie shows a simulation of the proneural wave model *Equations 17–19* simulated on a hexagonal lattice with circular geometry with a radius of 15 lattice sites. All simulation parameters are given in *Appendix 3—table 1*.
DOI: https://doi.org/10.7554/eLife.40919.007

For example, clones in which EGFR signalling has been constitutively activated tend to advance the transition zone within the clone, while the absence of EGFR signalling leads to loss of the proneural wave (*Figure 3a,b*) (*Yasugi et al., 2010*). To this end, we simulated the model dynamics on a two-dimensional hexagonal lattice that mimics the topology of the neuroepithelium. Consistent with experiment, the model captured the acceleration, delay or loss of the proneural wavefront within a clone depending upon its genetic makeup (*Figure 3a–f*; Appendix 6).

Low levels of Notch signalling activity are observed both in the neuroepithelium and in neuroblasts but not at the transition zone (*Figure 1c*). It has been shown experimentally that Notch activity is required to maintain neuroepithelial cell fate and the loss of Notch results in premature transformation into neuroblasts (*Egger et al., 2010*; *Ngo et al., 2010*; *Reddy et al., 2010*; *Yasugi et al., 2010*; *Orihara-Ono et al., 2011*; *Wang et al., 2011b*; *Pérez-Gómez et al., 2013*). Intriguingly, despite the observation of active Notch signalling, there is no evidence of lateral inhibition in the neuroepithelium. Lateral inhibition causes neighbouring cells to acquire complementary cell fates and so results in the emergence of a 'salt-and-pepper pattern' of Notch signalling activity (*Bray, 2006*; *Shaya and Sprinzak, 2011*). The reason for the absence of salt-and-pepper Notch signalling in the neuroepithelium is not clear. Notably, our model predicts that the basal level of Notch activity observed in the the neuroepithelium could be the reason for the suppression of lateral inhibition patterns outside the transition zone. In our model, basal levels of Notch activity in the neuroepithelium lead to a spatially homogeneous 'oversaturation' that prevents the Delta levels from rising before being activated by EGFR signalling. This is the case even in the presence of biochemical fluctuations (*Figure 4a*). However, if basal Notch levels are lowered to small values compared to the threshold levels for activation and inhibition in our model, we indeed recapitulate the salt-and-pepper patterns that are a consequence of lateral inhibition (*Figure 4b*). An analytical argument for the suppression of lateral inhibitions through basal Notch activity is given in Appendix 5.

We tested this prediction of the model by lowering Notch levels in the neuroepithelium but we did not observe 'salt-and-pepper' patterns of Delta/Notch expression within clones expressing an RNAi against Notch (*Figure 4c*). However, the absence of the emergence of lateral inhibition is likely due to the complete loss of detectable Notch in cells expressing the RNAi (*Figure 4d*), while the reduction of Notch levels in the model prediction is more subtle (*Figure 4b*). Referring to the 'phase diagram' in *Figure 4e*, it can be seen that both basal and Delta-regulated Notch activity need to be in the appropriate range for lateral inhibition patterns to occur, which is difficult to achieve experimentally. Furthermore, our model entails that Notch downregulation is a necessary (but not generally sufficient) condition for inducing salt-and-pepper patterns.

## Dependence of wave speed and transition zone width on kinetic rate parameters

Next, we asked which aspects of the signalling and gene expression changes in our model have the largest effect on two important features of the system: the speed of the proneural wave and the width of the transition zone. To this end, we performed a sensitivity analysis on the kinetic rate parameters, as detailed in Appendix 7.

This analysis, based on the so-called 'Morris method' (*Morris, 1991*; *Campolongo et al., 2007*; *Wu et al., 2013*), entails a resampling of the parameter space of the model and yields three indices for each probed parameter. These indices indicate the impact of each parameter on the assessed output. The Morris indices $m$ and $m^*$ describe the impact of the respective parameter on the output, with $m$ including positive and negative effects (which may cancel each other as the parameter is varied) and $m^*$ the overall absolute effect (*Campolongo et al., 2007*). The third index $\sigma$ measures the non-linearity of the parameter/output relation and/or interactions with other parameters (*Wu et al., 2013*). Detailed definitions of the respective indices are given in Appendix 7.

Here we probed the effects of the kinetic rate parameters of EGFR signalling, Delta-Notch signalling and L'sc expression on the propagation speed of the proneural wave and the width of the transition zone. As expected, this analysis showed that the diffusion, gain and decay rate of EGFR signalling ($\eta$, $\mu_E$ and $k_E$) are the key regulators of wave speed, since EGFR signalling constitutes the driver of the wave in our model (*Figure 5a*). In contrast, L'sc gene expression (parametrised by $\mu_L$ and $k_L$) had almost no effect on wave speed since its dynamics is 'pulled' by the EGFR signalling front. Interestingly, the basal gain rate of Notch signalling ($\beta$) as well as its decay rate ($k_N$) play another prominent role in setting the wave speed. This is consistent with experimental data showing

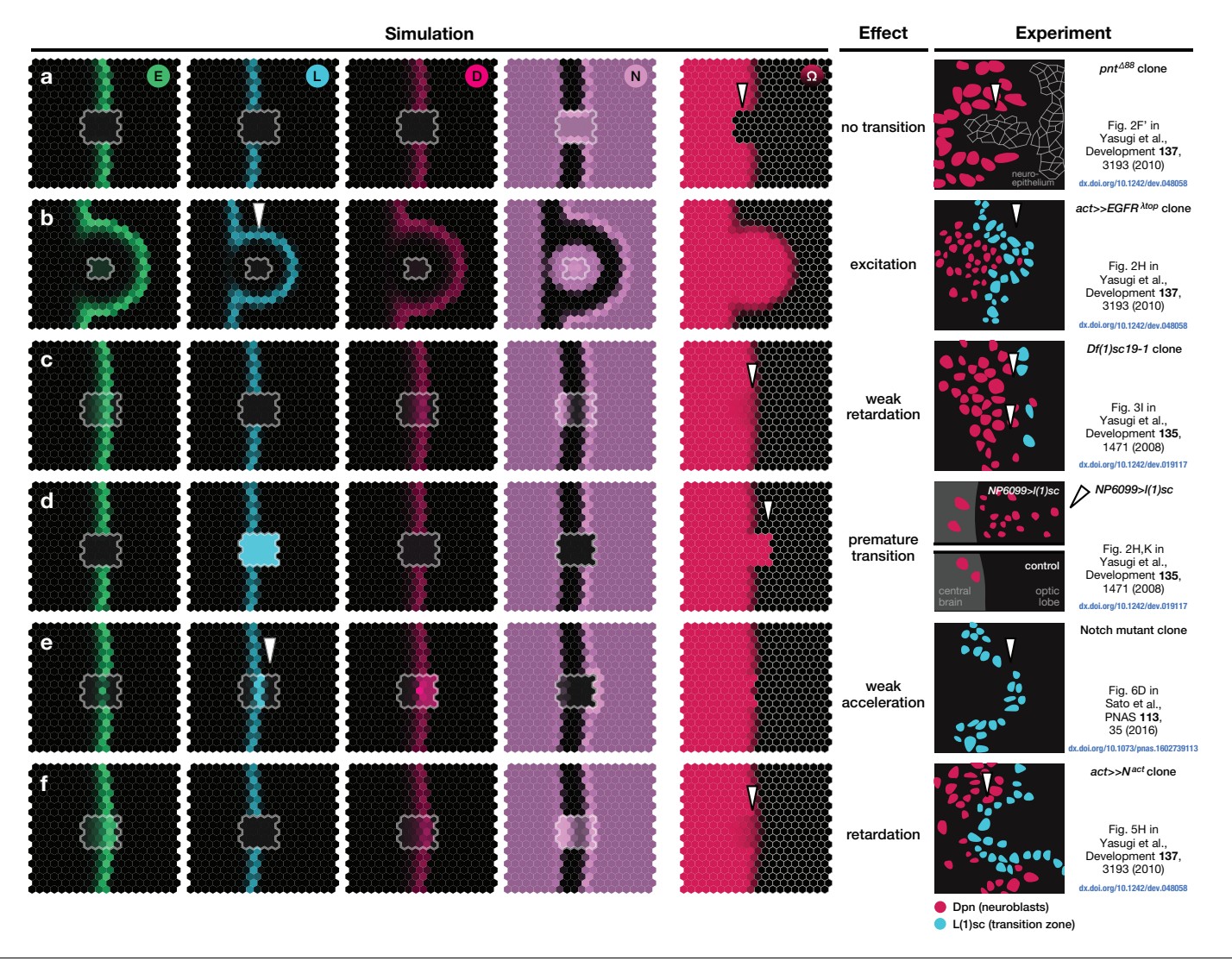

**Figure 3.** Simulations of the proneural wavefront as well as clones (outlined cells) capturing different mutant and transgenic conditions. (**a**) Knockout of EGFR signalling ($\mu_E = 0$ within the clone), (**b**) EGFR signalling constitutively active (E signalling always active within the clone), (**c**) L'sc knockout ($\mu_L = 0$ within the clone), (**d**) L'sc constitutively active (L synthesis always active within the clone), (**e**) Notch downregulation ($\beta = 0$ within the clone), (**f**) Notch upregulation (additional N synthesis with rate $\beta/2$ within the clone). In all panels, white arrowheads indicate advancements and retardations of the wavefront as compared to wildtype tissue due to the respective genetic alterations of the clones. The system given by **Equations 17–19** was numerically simulated on a $20 \times 20$ hexagonal lattice with initially localised levels of E in the first three columns at the left boundary of the system so that the wave travels to the right. All other parameters are given in **Appendix 3—table 1**. All shown simulation snapshots are taken at time $t = 25$, except for panel B, which is taken at $t = 17.5$. The column 'Experiment' shows sketches of experiments with mutant and transgenic clones and animals and refer to the corresponding original literature.
DOI: https://doi.org/10.7554/eLife.40919.008

that Notch signalling promotes EGFR signalling at the transition zone, leading to a reinforcement of activation as the wave arrives at undifferentiated cells (*Yasugi et al., 2010*). Considering the width of the transition zone (*Figure 5b*), we found that while all parameters had some effect, L'sc expression clearly had a tightening effect as it promoted differentiation and therefore leads to a faster termination of differentiation. In contrast, basal Notch signalling in the epithelium (described by $\beta$ and $k_N$) tended to enlarge the width of the transition zone in our sensitivity analysis. Indeed, a recent study showed that overexpression of Notch at the transition zone extends the width of the L'sc stripe and delays the transformation into neuroblasts (*Contreras et al., 2018*), providing support for

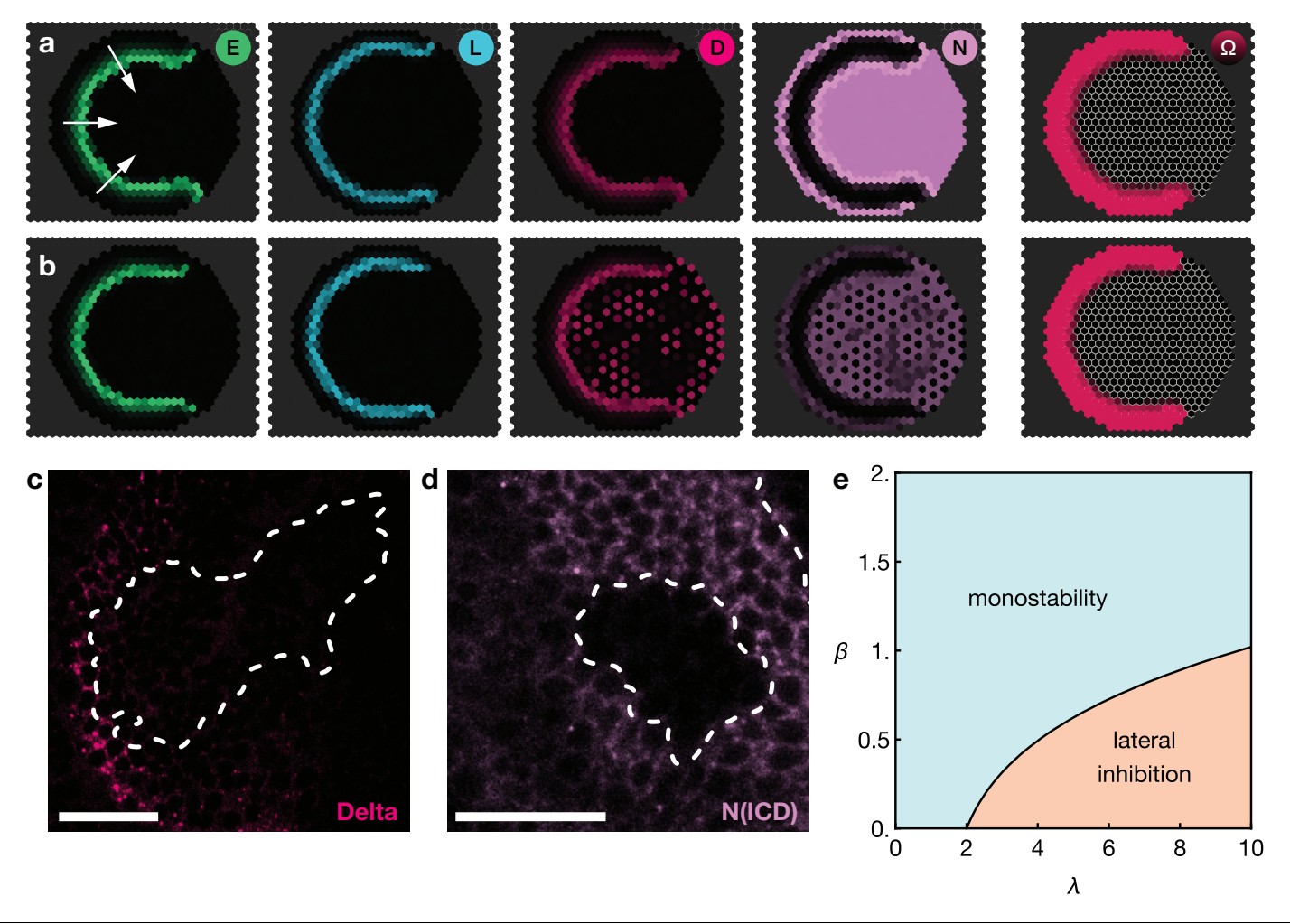

**Figure 4.** Basal Notch activity suppresses lateral inhibition patterns. The panels show snapshots of the proneural wave model *Equation 17–Equation 20* simulated on a hexagonal lattice with circular geometry with a radius of 15 lattice sites. White arrows indicate the direction of wave progression. Transient lateral inhibition patterns can occur if basal Notch levels are low compared to the thresholds for activation and inhibition of the Delta–Notch interactions: (a) In the scenario with basal Notch activity, lateral inhibition patterns are suppressed (basal Notch gain rate $\beta = 10$). (b) In the scenario with downregulated basal Notch activity, lateral inhibition patterns appear (basal Notch gain rate $\beta = 1$). Other parameters are given in *Appendix 3—table 1*; both panels are simulated with biochemical noise strength $\gamma/\mu_E = 0.5$ (see *Equation 20*). Initial conditions were localised elevated levels of E in those outer boundary cells that have angles between $\pi/3$ and $5\pi/3$ as measured from the center of the circular lattice. (c) Downregulation of Notch levels by expressing Notch RNAi in clones does not result in the emergence of a salt-and-pepper expression pattern of Delta (pink). Clone outlines are marked by white dotted lines. (d) Expressing Notch RNAi in clones results in the complete loss of detectable Notch (N (intracellular domain, ICD), purple) within the clones. Clone outlines are marked by white dotted lines. (e) Phase diagram for the occurrence of lateral inhibition in the two-cell system (for details, see Appendix 5). Here, $\beta$ denotes the basal production rate and $\lambda$ denotes the gain rate. (c) and (d) are single section confocal images, scale bars represent 20 $\mu$m.

DOI: https://doi.org/10.7554/eLife.40919.009

this prediction of the model. Subsequently, it follows that a complementary prediction of the model would be that a reduction of Notch at the transition zone would decrease the width of the L'sc stripe. We tested this prediction by knocking down Notch at the transition zone in clones (*Figure 6*). Within clones expressing Notch RNAi, the proneural wave was not only accelerated (*Figure 3*), as observed previously in Notch mutant clones (*Egger et al., 2010*; *Reddy et al., 2010*; *Yasugi et al., 2010*), but the width of the transition zone also appeared smaller (yellow arrowheads in *Figure 6*). In summary, this sensitivity analysis suggests that EGFR and Notch signalling are the key regulators of wave speed and width of the transition zone while L'sc expression provides an additional

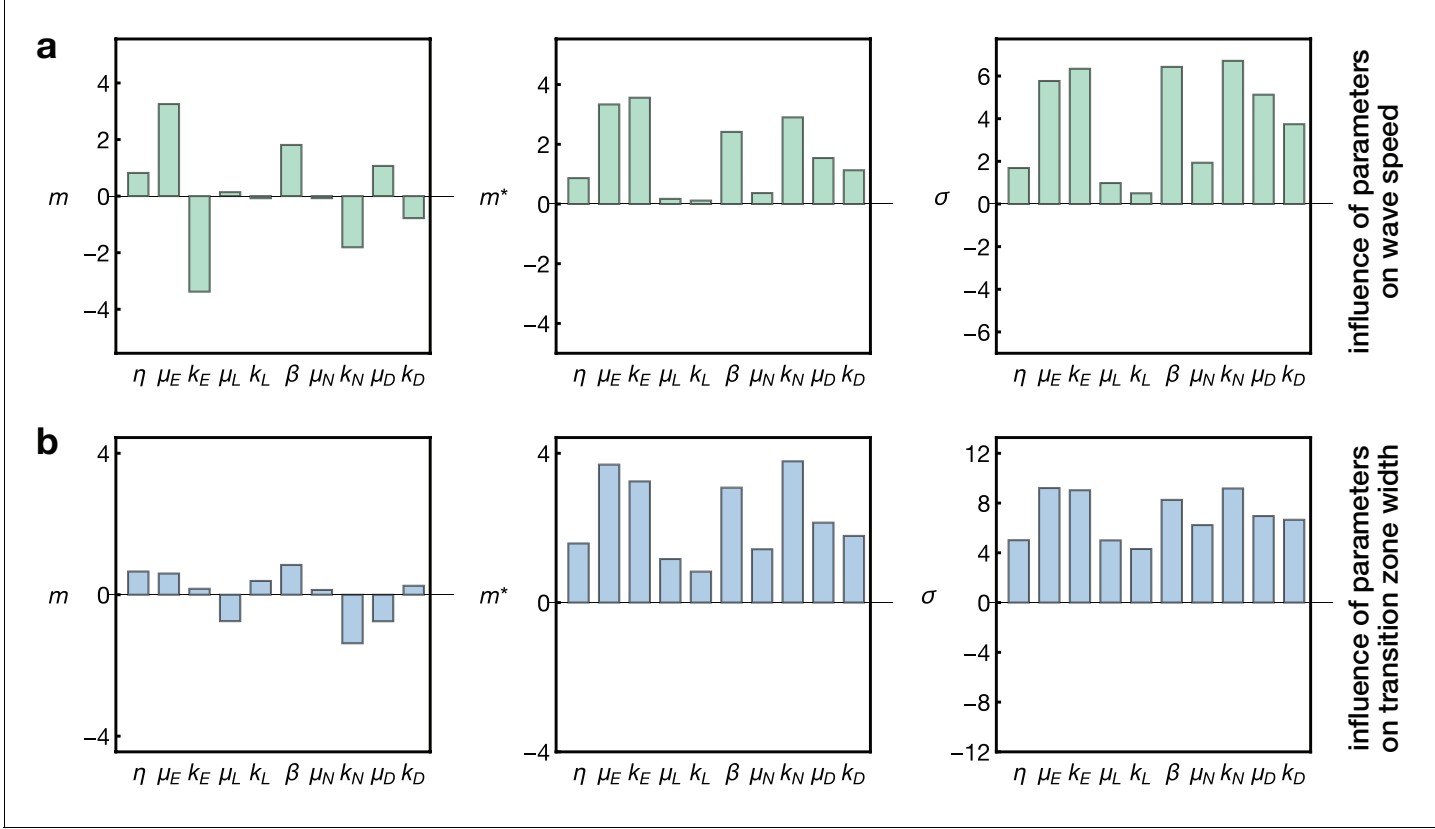

**Figure 5.** Sensitivity analysis of the model. Plots show the Morris indices $m$, $m^*$ and $\sigma$ as described in the main text and Appendix 7, indicating the effect of a parameter on (a) the wave speed and (b) the width of the transition zone. The indices $m$ and $m^*$ indicate the influence of a parameter on the respective output with $m$ comprising positive and negative effects and $m^*$ measuring the absolute effect, whereas non-zero values of $\sigma$ indicate a nonlinear influence and/or interactions with other parameters. The $\mu_i$ and $k_i$ denote the gain and decay rates for the respective components $i = \mathrm{E}, \mathrm{L}, \mathrm{D}, \mathrm{N}$, $\eta$ denotes the diffusion constant of the component 'E' and $\beta$ denotes the basal Notch gain rate (see *Equation 17*).
DOI: https://doi.org/10.7554/eLife.40919.010

acceleration of the transition of the wave and therefore has a negative influence on the width of the transition zone.

## Ectopic excitation of the transition in vivo

A signature feature of the model dynamics is that, through interactions between L'sc and EGFR signalling, the neuroepithelium functions as an excitable medium. As such, the model predicts that local induction of EGFR signalling would initiate a circular (target-like) transition wave (*Figure 7a*; *Video 5*). To test whether a transition wave in the neuroepithelium could be excited at a position remote from the proneural wave, we induced clones expressing a downstream effector of the EGFR signalling pathway, Pointed P1 (PntP1; *Figure 7b,c*; Materials and methods). We found upregulation of L'sc at clonal boundaries and expression of Dpn within the clone, suggesting the ectopic generation of neuroblasts within the epithelium, that is, a NE to NB transition (*Figure 7b,c*). Our results agree with previous experiments showing the induction of neuroblasts within the neuroepithelium in response to ectopic EGFR signalling (*Yasugi et al., 2010*) and are in striking agreement with model simulations based on the same perturbation (*Figure 7a*; *Video 5*; Supplementary Text).

## Discussion

Our findings suggest that the proneural wave involves the activation of an excitatory pulse of signalling activity and gene expression, giving rise to a tightly-regulated propagating transition zone. In contrast with Turing-based activator-inhibitor mechanisms (*Turing, 1952*), which typically comprise

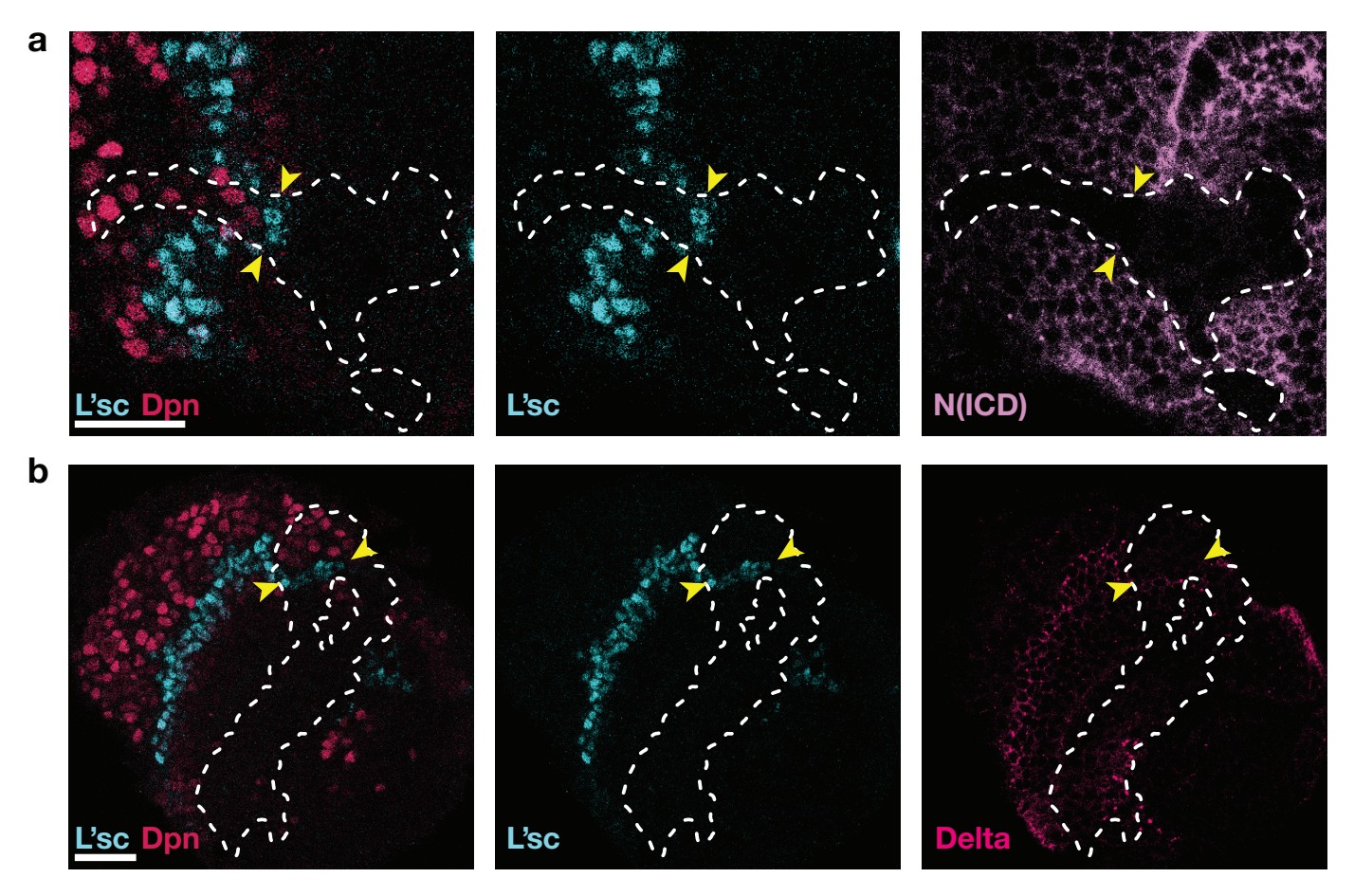

**Figure 6.** As in Notch mutant clones (*Egger et al., 2010*; *Reddy et al., 2010*; *Yasugi et al., 2010*), the proneural wave is accelerated in clones expressing Notch RNAi, see also *Figure 3e*. (a) Expression of Notch RNAi resulted in the downregulation of Notch (N(ICD), purple) and accelerated the transformation (which requires L'sc, cyan) of neuroepithelial cells to neuroblasts (Dpn, red) within clones. (b) The downregulation of Notch appears to decrease the width of the transition zone, as assessed by L'sc (cyan) and Delta (pink) expression within Notch RNAi clones. Dotted white lines mark clone boundaries and yellow arrows indicate the position of the transition zone within Notch RNAi clones. Images are single section confocal slices, scale bars represent 20 $\mu$m.

DOI: https://doi.org/10.7554/eLife.40919.011

fast diffusible inhibitors, the reaction-diffusion system described here is based on strictly local inhibition (*Figure 2b,c*). The role of sequential patterning by the proneural wave is to ensure the correct timing and composition of the neuroblast population (*Bertet et al., 2017*). A similar process of sequential patterning occurs during the progression of the morphogenetic furrow in the *Drosophila* eye (*Roignant and Treisman, 2009*; *Lubensky et al., 2011*; *Formosa-Jordan et al., 2012*; *Wartlick et al., 2014*; *Fried et al., 2016*; *Gavish et al., 2016*). However, the progression of the morphogenetic furrow also entails transient growth as well as subsequent photoreceptor patterning and differentiation to generate ommatidia (*Sato et al., 2013*).

In our model, which focuses on the driving mechanism behind the proneural wave, we have refrained from considering additional signalling pathways that neither play key roles in driving the proneural wave nor exhibit strong signatures of bidirectional feedbacks (in contrast to EGFR and Delta-Notch signalling). These include the JAK/STAT and Hippo pathways, which serve important roles in modulating proneural wave progression but are not actively involved in propagating the transition zone through a reaction-diffusion-like mechanism (*Yasugi et al., 2008*; *Reddy et al., 2010*; *Yasugi et al., 2010*; *Kawamori et al., 2011*; *Wang et al., 2011a*; *Weng et al., 2012*).

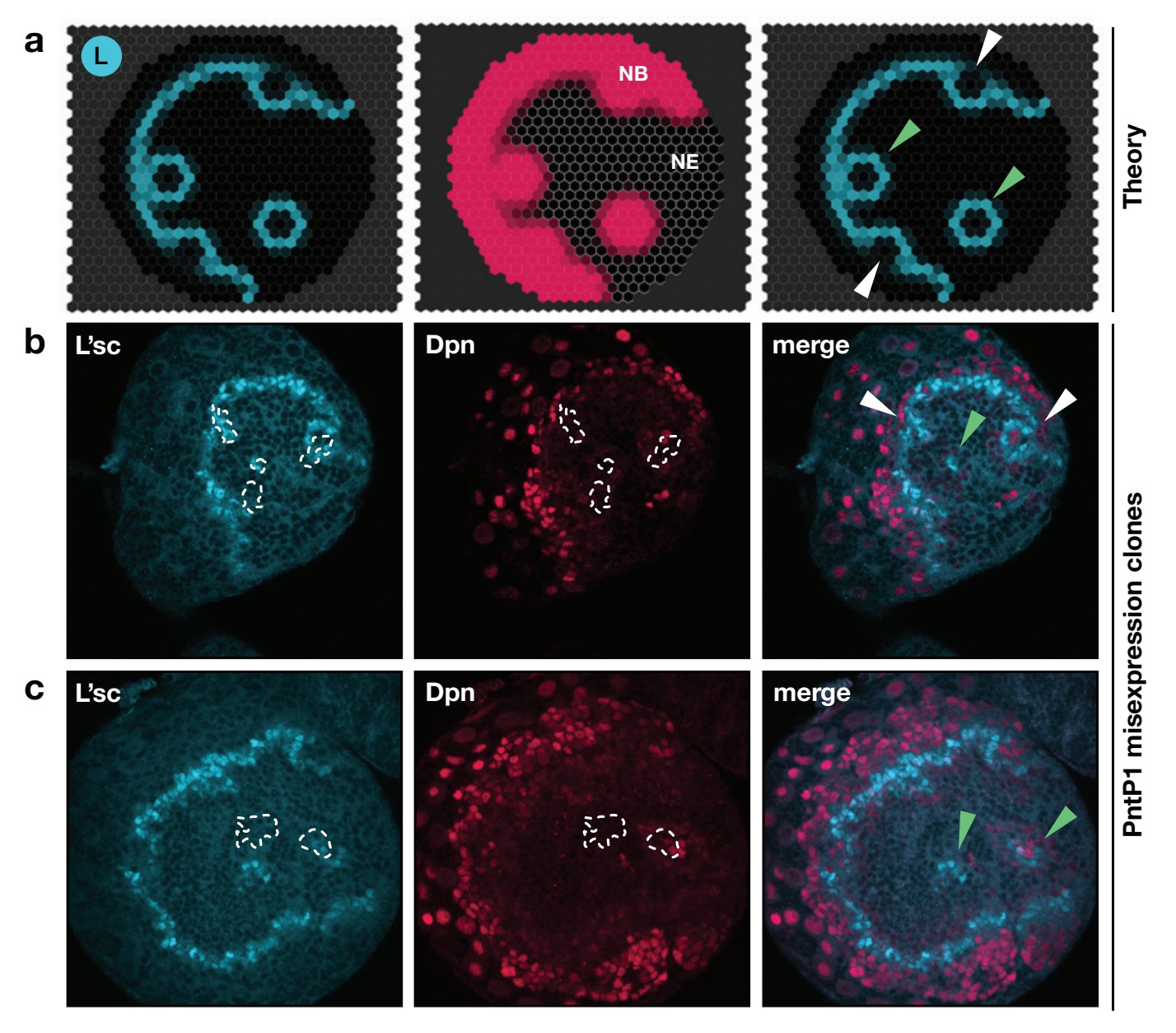

**Figure 7.** Integrated model of EGFR signalling, L'sc expression, Delta-Notch signalling and the NE to NB transition predicts key features of proneural wave progression in wildtype tissue and following perturbation. The effect of constitutively active EGFR signalling outside the transition zone: Comparison between the model prediction and experiment. (a) Snapshots of a model simulation on a two-dimensional hexagonal lattice representing the neuroepithelium with randomly distributed clones derived as target waves centred on the site (cell) in which EGFR signalling has been activated (see Materials and methods and *Video 5*). Cyan indicates levels of L'sc, red indicates neuroblasts (NB) and the grey grid the neuroepithelium (NE). The third column shows a merged image. (b,c) Ectopic expression of PntP1 within the neuroepithelium induces L'sc expression and a NE to NB transition. Clones expressing PntP1, a downstream effector of the EGFR signalling pathway, are indicated by white outlines or green arrowheads. Clones that merge with the transition zone are marked with white arrowheads. Clones within the neuroepithelium that are clearly separated from the transition zone, are marked by green arrowheads; clones that merge with the transition zone are marked with white arrowheads. L'sc is labelled in cyan; neuroblasts are labelled in red by expression of the Hes family transcription factor Deadpan (Dpn).
DOI: https://doi.org/10.7554/eLife.40919.013

**Video 5.** Travelling proneural wave with ectopic activation of EGFR signalling within clones. The movie shows a simulation of the proneural wave model *Equations 17–19* on the same lattice as in *Video 3* but with four single-cell clones with constitutively active EGFR signalling. The movie corresponds to the snapshots shown in *Figure 7a*.
DOI: https://doi.org/10.7554/eLife.40919.012

On a mechanistic level, the excitable propagation behaviour illuminated here provides a mechanism to capture the transient and localised activity of the proneural gene *l'sc* and EGFR signalling, as well as robustness against fluctuating signalling activity and gene expression. In contrast to a recent model (*Sato et al., 2016*), our model also implies that neither differentiation nor proneural gene expression is required for the transient stabilisation of EGFR signalling activity that is required to advance the wave front. In the context of vertebrate somitogenesis, intracellular excitability has recently been suggested to underlie the emergence of genetic oscillations (*Hubaud et al., 2017*). The appearance here of excitability in the context of a propagating front of gene expression suggests that such a mechanism may serve more widely as a generic and robust strategy to achieve sequential transition waves in developing tissues.

## Materials and methods

### Fly strains
Flies were raised on standard cornmeal medium at 25°C. Strains used were:
 yw, hsFLP; FRT40A, tub-GAL80/CyO, ActGFP; tubP-GAL4,UAS-mCD8-GFP/TM6B
 w; FRT40A; UAS-Pnt-P1/TM6B
 w; FRT40A/CyO; N RNAi/TM6B (N RNAi lines used were BL33611 and BL33616)
 HLH-mgamma-GFP (*Almeida and Bray, 2005*) was used to report active Notch signalling.

### UAS-Pnt-P1 clones and N RNAi clones
To generate clones in the developing optic lobe, larvae were collected 48–50 hr after egg laying and were heat shocked for 20 min at 37°C. Larvae were then dissected 50 or 60 hr after clone induction.

### Immunohistochemistry
Larval brains were dissected in PBS and fixed for 20 min at room temperature in 4% formaldehyde and fixation buffer (PBS, 5 mM MgCl2, 0.5 mM EGTA). After fixation, brains were rinsed and washed in 0.3% PBS Triton X100 (PBT). Samples were blocked with 10% normal goat serum (NGS) in 0.3% PBT at room temperature and incubated with the primary antibodies overnight at 4°C. Brains were then washed in 0.3% PBT and incubated with the secondary antibodies overnight at 4°C. Brains were washed in 0.3% PBT and mounted in Vectashield (Vector Laboratories, Burlingame, CA, USA). The following primary antibodies and dilutions were used: guinea pig anti-Dpn (1:10,000) and rat anti-L'sc (1:5,000) (*Caygill and Brand, 2017*), chicken anti-GFP (1:2,000) from Abcam, mouse anti-Delta (1:50, C594.9B) from DSHB and mouse anti-Notch (intracellular domain, ICD) (1:50, C17.9C6) from DSHB. Fluorescently conjugated secondary antibodies Alexa405, Alexa488, Alexa546 and Alexa633 (all 1:200) from Life Technologies.

Images were acquired with a Leica TCS SP8 confocal microscope (Leica Microsystems, Wetzlar, Germany) and analysed with Fiji (*Schindelin et al., 2012*). Figures and illustrations were assembled using Adobe Photoshop CS3 and Adobe Illustrator CS3 (Adobe Systems, San Jose, CA, USA).

## Acknowledgments

We are grateful to Pau Formosa-Jordan, Claude Desplan, François Schweisguth and members of the Simons and Brand labs for useful discussions.

## Additional information

### Funding

| Funder | Grant reference number | Author |
|---|---|---|
| Wellcome | 092096 | David J Jörg<br>Elizabeth E Caygill<br>Anna E Hakes |
| Cancer Research UK | C6946/A14492 | David J Jörg<br>Elizabeth E Caygill<br>Anna E Hakes<br>Esteban G Contreras<br>Andrea H Brand<br>Benjamin D Simons |
| Biotechnology and Biological Sciences Research Council | Project Grant BB/L007800/1 | Elizabeth E Caygill<br>Andrea H Brand |
| Wellcome | Senior Investigator Award 103792 | Anna E Hakes<br>Esteban G Contreras<br>Andrea H Brand |
| Royal Society | Darwin Trust Research Professorship | Andrea H Brand |
| Wellcome | Senior Investigator Award 098357 | Benjamin D Simons |
| Royal Society | EP Abraham Research Professorship RP\R1\180165 | Benjamin D Simons |

The funders had no role in study design, data collection and interpretation, or the decision to submit the work for publication.

### Author contributions

David J Jörg, Conceptualization, Software, Formal analysis, Investigation, Visualization, Methodology, Writing—original draft, Writing—review and editing; Elizabeth E Caygill, Investigation, Visualization, Methodology, Writing—original draft, Writing—review and editing; Anna E Hakes, Esteban G Contreras, Investigation, Visualization, Methodology, Writing—review and editing; Andrea H Brand, Benjamin D Simons, Conceptualization, Supervision, Funding acquisition, Investigation, Methodology, Writing—original draft, Project administration, Writing—review and editing

### Author ORCIDs

David J Jörg (iD) http://orcid.org/0000-0001-5960-0260
Anna E Hakes (iD) http://orcid.org/0000-0002-8664-1014
Esteban G Contreras (iD) http://orcid.org/0000-0002-0934-741X
Andrea H Brand (iD) https://orcid.org/0000-0002-2089-6954
Benjamin D Simons (iD) http://orcid.org/0000-0002-3875-7071

### Decision letter and Author response

Decision letter https://doi.org/10.7554/eLife.40919.030
Author response https://doi.org/10.7554/eLife.40919.031

## Additional files

### Supplementary files

• Transparent reporting form
DOI: https://doi.org/10.7554/eLife.40919.014

### Data availability

All data generated or analysed during this study are included in the manuscript.

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

## Appendix 1

DOI: https://doi.org/10.7554/eLife.40919.015

## Bistable EGFR signalling fronts

We describe the details of our theoretical model by building up an integrated model of the proneural wave starting from a simple system of EGFR signalling activity. Reaction-diffusion models with different types of genetic interactions and correspondingly different phenomenologies have been used to describe the dynamics of EGFR signalling in other contexts such as *Drosophila* oogenesis (**Shvartsman et al., 2002**; **Zartman et al., 2009**; **Simakov et al., 2012**; **Fauré et al., 2014**) and the *Drosophila* retina (**Pennington and Lubensky, 2010**).

### Model formulation

To illustrate the mechanism of wave propagation, we first show how EGFR signalling activity alone may give rise to a propagating signalling front. As outlined in the main text, we consider the dynamics of one signalling component E, which encapsulates the collective dynamics of the EGFR/Rhomboid/Spitz network (see below) and, hence, involves positive feedback and effective diffusion on the tissue level (see *Figure 1c* and *Figure 2a*). We describe the component E by a continuous field that defines the signalling activity, $\phi = \phi(\mathbf{x}, t)$, where $\mathbf{x}$ denotes the position coordinate within the tissue and $t$ the time. (Throughout this text, we consider one-dimensional systems for illustration purposes and two-dimensional systems to describe the proneural wave *in vivo*. Hence, the position variable $\mathbf{x}$ and the nabla operator $\nabla$ refer to $\mathbf{x} = (x_1, x_2)$ and $\nabla = (\partial/\partial x_1, \partial/\partial x_2)$ in the case of two dimensions and $\mathbf{x} = x$ and $\nabla = \partial/\partial x$ in the case of one dimension.) The corresponding reaction-diffusion system is given by

$$\frac{\partial \phi}{\partial t} = \eta \nabla^2 \phi + r(\phi) , \qquad (3)$$

where $\eta$ is the effective diffusion constant and $r(\phi)$ is the reaction term describing the intracellular feedback,

$$r(\phi) = \mu h\left(\frac{\phi}{\Phi}\right) - k\phi . \qquad (4)$$

Here, $\mu$ denotes the gain rate in signalling activity due to positive feedback, $k$ the degradation rate and $h$ is a monotonically increasing function describing the nonlinear positive feedback with $\Phi$ being the threshold activity. Here, we choose a function of the Hill type (**Novák and Tyson, 2008**),

$$h(\phi) = \frac{\phi^n}{1 + \phi^n} , \qquad (5)$$

where $n$ is the 'Hill exponent' characterising the nonlinearity of the feedback. Systems of the type given by *Equations 3–5* are known to exhibit solutions in the form of a propagating front for appropriate parameters (**Keener and Sneyd, 2009**). For illustrative purposes, we initially consider the dynamics of signalling activity in one spatial dimension on a domain of length $\ell$ with 'no-flux' boundary conditions, $(\partial \phi / \partial x)|_{x=0} = 0 = (\partial \phi / \partial x)|_{x=\ell}$. A numerical example of the system specified by *Equations 3–5* is shown in *Figure 2a*. A localised concentration beyond a certain threshold concentration at the boundary $x = 0$ initiates a travelling front of E, which propagates at constant velocity, leaving behind elevated levels of sustained signalling activity. (For the example shown in *Figure 2a*, we choose an initial condition of the form $\phi(x, 0) = e^{-x^2/x_0^2}$ with $x_0^2 = 5\eta/k$.)

This behaviour is familiar in reaction-diffusion systems with reaction terms of the form given by *Equation 4* and can be understood by studying the local reaction dynamics in a phase portrait: *Appendix 1—figure 1a* shows the reaction term $r$, which describes the local net loss/

gain rate of signalling activity E. For sufficiently large gain rates, the function has three equilibrium points $\phi_i^*$ for which the net loss/gain rate vanishes (dots in **Appendix 1—figure 1a**). Therefore, once the system has reached such an equilibrium, it remains there until perturbed. The two equilibrium points $\phi_0^*$ and $\phi_2^*$ are stable (black dots), that is small perturbations of the signalling activity will drive the system back to these equilibrium points— higher levels lead to a net loss whereas smaller levels lead to a net gain (black arrows in **Appendix 1—figure 1a**). The two equilibria are separated by an unstable equilibrium $\phi_1^*$ (white dot). If the system is only slightly perturbed around $\phi = \phi_0^*$ (black dot), the reaction dynamics will drive the system back towards this equilibrium, because degradation is stronger than self-activation in this region, $r < 0$. However, if the signalling activity rises above $\phi_1^*$ (e.g., due to diffusion from the neighboring cell), self-activation will outcompete degradation, $r > 0$, and the signalling activity will be driven towards the other stable equilibrium $\phi_2^*$. This reaction-diffusion mechanism is known as a bistable front (**Keener and Sneyd, 2009**).

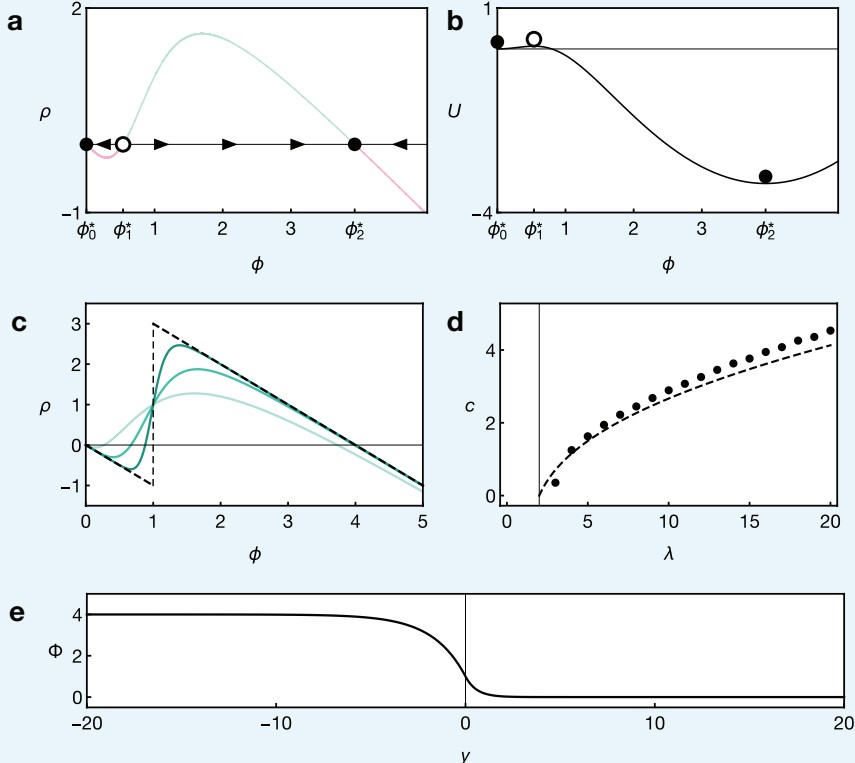

**Appendix 1—figure 1.** Key features of reaction dynamics leading to bistable front propagation. (**a**) Reaction term $\rho$ as given by **Equation 6**. Dots indicate the fixed points $\phi_i^*$ with $i = 0, 1, 2$ for which $\rho(\phi_i^*) = 0$. Filled dots indicate stable fixed points, the open dot indicates the unstable fixed point. (**b**) Potential $U$ associated with the reaction term $\rho$ shown in panel A and defined by $\rho = -\partial U / \partial \phi$. Parameters in both panels are $\lambda = 4$ and $n = 3$. (**c**) Reaction term $\rho$ as given by **Equation 6** for Hill exponents $n = 2$ (light green), $n = 4$ (green), $n = 10$ (dark green), and the limiting case $n \to \infty$ (dashed black), given by **Equation 9**, for $\lambda = 4$. (**d**) Front velocity $c$ as a function of $\lambda$. Numerical solutions obtained from simulations of **Equation 6** with Hill exponent $n = 2$ and analytical approximation **Equation 11** for the $n \to \infty$ limit. (e) Example of the front profile $\phi$ given by **Equation 10** with $\lambda = 4$.
DOI: https://doi.org/10.7554/eLife.40919.016

Further insight into the dynamics of the front propagation can be gained using analytical techniques (**Keener and Sneyd, 2009**). Non-dimensionalising the model by rescaling $\phi \to \phi / \Phi$, $t \to kt$, and $x \to \sqrt{k/\eta}x$, the corresponding reaction-diffusion system in one dimension is given by

$$\frac{\partial \phi}{\partial t} = \frac{\partial^2 \phi}{\partial x^2} + \rho(\phi) , \qquad \rho(\phi) = \lambda h(\phi) - \phi .  \tag{6}$$

Here, $\lambda = \mu/(k\Phi)$ is the only (dimensionless) parameter and $\rho$ is the dimensionless reaction term.

Anticipating that the system gives rise to a travelling front with velocity $c$ and a stationary front profile $\phi$, we make the ansatz $\phi(x,t) = \phi(x - ct)$ in **Equation 6**, which yields the ordinary differential equation

$$\phi'' + c\phi' + \rho(\phi) = 0 ,  \tag{7}$$

where the prime denotes the derivative with respect to the argument $y = x - ct$ of $\phi$.

Multiplying this equation by $\phi'$ and integrating from $-\infty$ to $\infty$ yields an implicit expression for the velocity of the front,

$$c = \frac{\Delta U}{\int (\phi')^2 \, \mathrm{d}y} ,  \tag{8}$$

where $\Delta U = \lim_{y \to \infty}[U(\phi(y)) - U(\phi(-y))]$ with the potential $U(\phi)$ defined by $\rho = -\partial U/\partial \phi$ (see **Appendix 1—figure 1b**). Analytical solutions to **Equation 8** only exist for special functional forms of the reaction term $\rho$. To obtain analytical insights into the wave speed, we consider the limiting case of large Hill exponents, $n \to \infty$ in **Equation 5**. This corresponds to a switch-like response of the gain rate once the activation threshold is reached. In this case, the reaction term acquires a piecewise linear functional form,

$$\rho(\phi) = \lambda \Theta(\phi - 1) - \phi ,  \tag{9}$$

where $\Theta$ is the Heaviside step function with the convention $\Theta(0) = 1/2$ (see **Appendix 1—figure 1c**). This function has three equilibria for $\lambda > 1$. For this reaction term, the front profile and front velocity can be calculated analytically via **Equation 7** and **Equation 8**. The front profile is given by

$$\phi(y) = \begin{cases} e^{-\sqrt{\lambda-1}\, y} & y \geq 0 \\ \lambda + (1-\lambda)e^{y/\sqrt{\lambda-1}} & y < 0 \end{cases} ,  \tag{10}$$

where we have fixed the arbitrary position of the co-moving reference frame by imposing the condition $\phi(0) = 1$. **Appendix 1—figure 1e** shows an example of the front solution **Equation 10**. The velocity of the front is given by

$$c = \frac{\lambda - 2}{\sqrt{\lambda - 1}} .  \tag{11}$$

Expanding in the limit of large $\lambda$, dropping all orders higher than $\lambda^{-1/2}$ and restoring the original parameter dependence by multiplying with the velocity scale $\sqrt{\eta k}$, we find

$$c \approx \sqrt{\frac{\eta\mu}{\Phi}} - \frac{3}{2}\sqrt{\frac{\eta\Phi}{\mu}} k .  \tag{12}$$

**Equation 12** implies that the front velocity increases with increasing diffusion constant $\eta$ and synthesis rate $\mu$ as well as decreasing degradation rate $k$ and decreasing activation threshold $\Phi$. **Appendix 1—figure 1d** shows that **Equation 11** is a good approximation for the velocity of the front even when compared to numerical simulations with a finite Hill exponent $n$.

## A one-component model reproduces the key features of more detailed models

Previously, we sought to capture the spatio-temporal dynamics of EGFR signalling by a single component, described by the signalling activity $\phi$, even though EGFR signalling comprises three components: the EGF receptor, the transmembrane factor Rhomboid and the EGFR ligand Spitz (see *Figure 1d*). To show that the system above entails the essential features of more detailed descriptions, we now consider the kinetics of these three components and show that a corresponding model gives rise to the same qualitative dynamics. We consider a dimensionally reduced reaction-diffusion system where $\psi^E$ denotes the concentration of *bound* EGFRs, $\psi^R$ represents the concentration of Rhomboid, and $\psi^S$ is the concentration of the *secreted* active form of Spitz (sSpi),

$$
\begin{aligned}
\frac{\partial \psi^E}{\partial t} &= \lambda_E H_E(\psi^S) - \psi^E \,, \\
\frac{\partial \psi^R}{\partial t} &= \lambda_R H_R(\psi^E) - \psi^R \,, \\
\frac{\partial \psi^S}{\partial t} &= \frac{\partial^2 \psi^S}{\partial x^2} + \lambda_S \psi^R - \psi^S \,.
\end{aligned}
\tag{13}
$$

Here, $\lambda_E$ is the binding rate of EGFRs, $\lambda_R$ is the synthesis rate of Rhomboid, and $\lambda_S$ is the secretion rate of Spitz; the functions $H_E$ and $H_R$ describe saturation of EGFR binding and saturation of the Rhomboid synthesis rate, respectively, and are assumed to be of the same qualitative form as the function $h$, given by *Equation 5*. Only the equation for the component S contains a diffusion term as only Spitz is secreted. The negative terms account for receptor unbinding and degradation of gene products; for simplicity, we have chosen identical unit decay rates and concentration thresholds. Numerical examples of *Equation 13* show that the positive feedback of E, R, and S can generate a bistable front, see *Appendix 1—figure 2*.

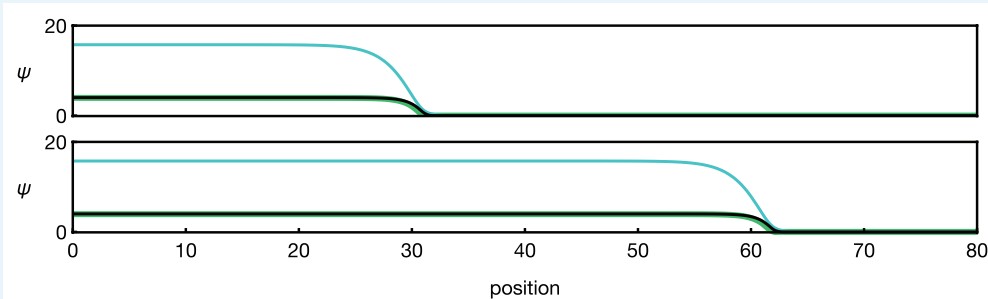

**Appendix 1—figure 2.** Numerical example of the three-component system *Equation 13*. Different curves show $\psi^E$ (black), $\psi^R$ (green), and $\psi^S$ (blue). The two panels show the time points $t = 30$ (top) and $t = 60$ (bottom). Functions and parameters are $h_R(\phi) = h_E(\phi) = h(\phi)$, given by *Equation 5* with $n = 3$, and $\lambda_i = 4$ for $i = E, R, S$. Boundary conditions are $(\partial \psi^i / \partial x)|_{x=0} = 0 = (\partial \psi^i / \partial x)|_{x=\ell}$.
DOI: https://doi.org/10.7554/eLife.40919.017

Since each component is required to activate the next one in the loop, all three components show the same behaviour in terms of their localisation. Moreover, since the activation of one component is sufficient to activate the entire loop, diffusibility of S confers a diffusion-like effect to the entire E–R–S system. Therefore, the parameter $\mu$ of the one-component model *Equation 3* and *Equation 4* is to be interpreted as a composite parameter that characterises the strength of the positive feedback of the entire cycle. Likewise, the parameter $k$ represents the combined effects of degradation and receptor unbinding, and $\eta$ describes the effective diffusibility, mediated by diffusion of component S. Formally, a connection to the one-component model *Equation 6* can be established when the dynamics of E and R is sufficiently fast such that they are always close to their stationary values

determined by $\partial \psi^{\mathrm{E}}/\partial t = 0$ and $\partial \psi^{\mathrm{R}}/\partial t = 0$ while the front is travelling. In this case, we can solve for $\psi^{\mathrm{E}}$ and $\psi^{\mathrm{R}}$ and obtain a closed equation for $\phi \equiv \psi^{\mathrm{S}}$,

$$\frac{\partial \phi}{\partial t} = \frac{\partial^2 \phi}{\partial x^2} + \lambda h(\phi) - \phi \, , \tag{14}$$

where $\lambda \equiv \lambda_{\mathrm{S}}$ and $h(\phi) = \lambda_{\mathrm{R}} H_{\mathrm{R}}(\lambda_{\mathrm{E}} H_{\mathrm{E}}(\phi))$. Note that **Equation 14** is formally equivalent to **Equation 6**; if $H_{\mathrm{E}}$ and $H_{\mathrm{R}}$ are sigmoidal functions with the qualitive shape of **Equation 5**, then so is $h$.

# Appendix 2

DOI: https://doi.org/10.7554/eLife.40919.015

## Excitable dynamics from EGFR–proneural interactions

While the minimal model **Equations 3–5** leads to the emergence of a front that leaves behind an elevated signalling state, the transition zone of the proneural wave is characterised by localised EGFR signalling and proneural gene expression. We now show how such a travelling localised pulse arises if we take into account interactions between EGFR signalling and the proneural gene *l'sc*. The interactions between EGFR signalling and L'sc are schematically represented in **Figure 1f** and **Figure 2b**: the component E activates the expression of L while the component L effectively downregulates E as a consequence of the transition that is induced. The corresponding reaction-diffusion equations for the two fields $\phi^{\mathrm{E}}$ and $\phi^{\mathrm{L}}$ are given by

$$
\begin{aligned}
\frac{\partial \phi^{\mathrm{E}}}{\partial t} &= \nabla^2 \phi^{\mathrm{E}} + \rho_{\mathrm{E}}(\phi^{\mathrm{E}}, \phi^{\mathrm{L}}) \,, \\
\frac{\partial \phi^{\mathrm{L}}}{\partial t} &= \rho_{\mathrm{L}}(\phi^{\mathrm{E}}, \phi^{\mathrm{L}}) \,,
\end{aligned}
\tag{15}
$$

with the reaction terms

$$
\begin{aligned}
\rho_{\mathrm{E}} &= \lambda_{\mathrm{E}} h(\phi^{\mathrm{E}}) \bar{h}(\phi^{\mathrm{L}}) - \phi^{\mathrm{E}} \,, \\
\rho_{\mathrm{L}} &= \lambda_{\mathrm{L}} h(\phi^{\mathrm{E}}) - \phi^{\mathrm{L}} \,,
\end{aligned}
\tag{16}
$$

where we have introduced the Hill function $\bar{h}(\phi) \equiv 1 - h(\phi)$ which describes an inhibitory effect. For simplicity, we have also considered identical unit concentration thresholds for activation and inhibition. Here, $\lambda_{\mathrm{E}}$ and $\lambda_{\mathrm{L}}$ are the gain rates of the components E and L, respectively. Note that in the absence of L ($\phi^{\mathrm{L}} = 0$), the reaction term $\rho_{\mathrm{E}}$ reduces to the one of the one-component model given in **Equation 6**, $\rho_{\mathrm{E}}(\phi^{\mathrm{E}}, 0) = \rho(\phi^{\mathrm{E}})$ (**Appendix 2—figure 1**). Again, we consider a finite one-dimensional domain of length $\ell$ and no-flux boundary conditions, $(\partial \phi^{\mathrm{E}}/\partial x)|_{x=0} = 0 = (\partial \phi^{\mathrm{E}}/\partial x)|_{x=\ell}$ and $(\partial \phi^{\mathrm{L}}/\partial x)|_{x=0} = 0 = (\partial \phi^{\mathrm{L}}/\partial x)|_{x=\ell}$.

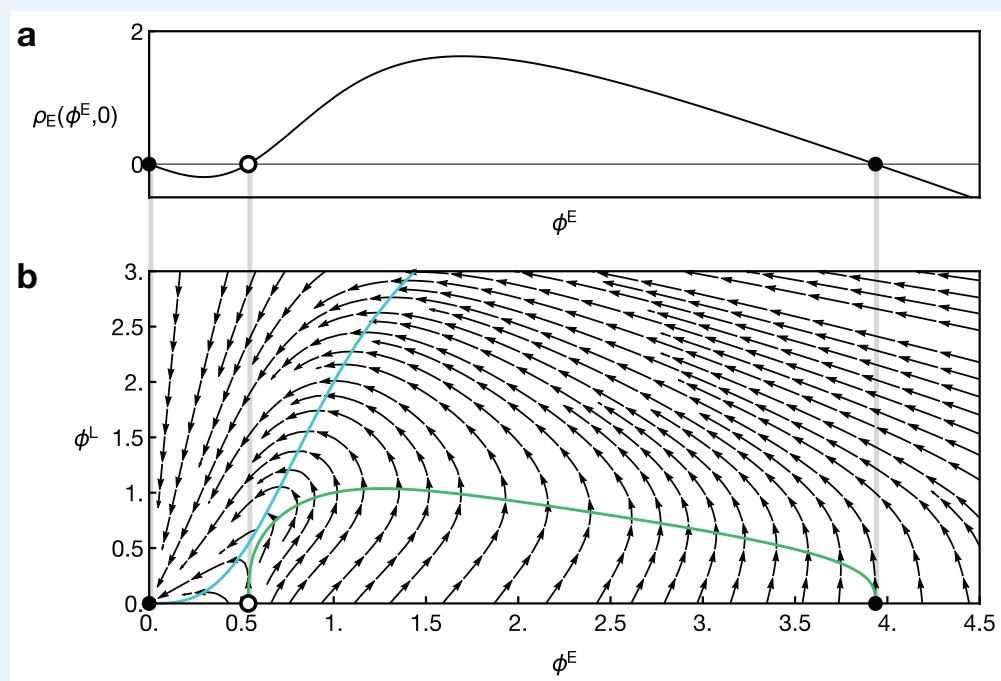

**Appendix 2—figure 1.** Reaction dynamics of the two-component model *Equation 15* and *Equation 16*. (**a**) Reaction term of the component E in the absence of L, given by $\rho_E(\phi^E, 0)$ in *Equation 16*. Dots indicate fixed points for which $\rho_E(\phi^E, 0) = 0$. (**b**) Full local reaction dynamics for the two-component model. Vector field $\mathbf{F} = (\rho_E, \rho_L)$ as given by *Equation 16*. Dots indicate points with $\phi^L = 0$ and $\rho_E = 0$. Parameters in both plots are $\lambda_E = \lambda_L = 4$ and $n = 3$. Coloured curves show the nullclines for E (green) and L (blue).
DOI: https://doi.org/10.7554/eLife.40919.019

*Figure 2b* in the main text displays a numerical example of the system *Equation 15* and *Equation 16*. Again, starting from a localised concentration of the component E at $x = 0$, a localised pulse of E and L travels through the system at a constant speed. (For the example shown in *Figure 2b*, we choose an initial condition of the form $\phi(x, 0) = e^{-x^2/x_0^2}$ with $x_0^2 = 10\eta/k$.) Again, this behaviour can be elucidated by studying the local reaction dynamics, which is now given by the vector field $\mathbf{F} = (\rho_E, \rho_L)$, defined by *Equation 16*) (*Appendix 2—figure 1*). Note that now only the point $(\phi^E, \phi^L) = (0, 0)$ is a stable equilibrium. The black and white dots indicate points with $\phi^L = 0$ that satisfy $\rho_E = 0$ as in the one-component system (compare to *Appendix 1—figure 1a*). However, note that only the point $(0, 0)$ also satisfies $\rho_L = 0$. Again, diffusion from a neighboring cell will increase the levels of E. When the concentration level marked by the white point is exceeded, the reaction dynamics will elevate the levels of E and L until a turning point is reached when downregulation of E is sufficiently strong to suppress its positive self-feedback. Thus, the system finally returns to the fixed point $(\phi^E, \phi^L) = (0, 0)$ which marks the end of the pulse. Hence, the model *Equation 15* and *Equation 16* leads to propagation of a pulse of E and L at a defined speed.

## Appendix 3

DOI: https://doi.org/10.7554/eLife.40919.015

# Integrated model of the proneural wave

The system described in Appendix 2 invokes a core mechanism giving rise to a propagating transition zone. As motivated in the main text, we now extend our model to include Delta–Notch interactions, based on the classical description advanced by (*Collier et al., 1996*) and also include cis-inhibition (*Sprinzak et al., 2010*; *Shaya and Sprinzak, 2011*).

An earlier attempt to describe the proneural wave advanced by (*Sato et al., 2016*) has focused on a phenomenological description of the proneural wave, which, already in its general approach, differs from our model. The corresponding model is, in major parts, built around observed patterns of gene expression and patterning phenomena, rather than starting from the molecular underpinnings: there, proneural gene expression is not an independent dynamic ingredient; rather, the cell state (NE or NB) and the state of proneural gene expression is combined in a single abstract variable, making it impossible to address the effects of alterations in proneural gene expression independently of the NE to NB transition. Hence, in (*Sato et al., 2016*), the general driving mechanism of the wave is fundamentally different from our model: it relies on EGFR signalling inducing a change in cell state and, conversely, a change in cell state transiently driving EGFR signalling activity through a phenomenological prescription. In contrast to our model, this renders autocrine EGFR signalling without intrinsic bistability and therefore without the ability to autonomously drive the wavefront. Moreover, the resulting model is unstable with respect to additive fluctuations in gene expression and signalling activity; slight misexpression or perturbations in signalling activity will result in an immediate premature differentiation, as discussed in (*Sato et al., 2016*). Delta-Notch interactions are incorporated as a subsystem without intrinsic multistability usually found necessary in attempts to describe the emergence of lateral inhibition phenomena (*Collier et al., 1996*).

Since Delta–Notch interactions can give rise to lateral inhibition, that is stable low-Delta/high-Notch and high-Delta/low-Notch states in adjacent cells, instead of a continuous description of the tissue, we now consider a lattice where each lattice site represents a cell. We consider the signalling and gene activities $\phi_{\mathrm{x}}^i(t)$ with $i = \mathrm{E, L, D, N}$ describing EGFR signalling activity, L'sc expression, and Delta and Notch, respectively. The index $\mathrm{x}$ indicates the lattice site. Furthermore, we introduce the cell state $\Omega_{\mathrm{x}}(t)$ which takes values from 0 to 1, where $\Omega_{\mathrm{x}} = 0$ indicates that cell $\mathrm{x}$ is a neuroepithelial cell and $\Omega_{\mathrm{x}} = 1$ indicates that cell $\mathrm{x}$ is a neuroblast. *Figure 2d* shows the regulatory network of the model. In contrast to the effective inhibition of signalling by the proneural gene *l'sc* that we considered before, we now include the more realistic shutdown of signalling as a consequence of differentiation. Moreover, motivated by the presence of low levels of Notch signalling in the neuroepithelium and the neuroblasts (*Egger et al., 2010*; *Orihara-Ono et al., 2011*), we include a basal source of Notch that is independent of trans-activation by Delta in adjacent cells. As in the previous sections, we here consider identical reference concentrations for activation and inhibition and rescale all concentrations by this reference concentration, so that the fields $\phi_{\mathrm{x}}^i$ with $i = \mathrm{E, L, D, N}$ are dimensionless. Moreover, we consider identical degradation constants for all four components and rescale time by the degradation time. The corresponding dynamical equations are given by

$$\frac{d\phi_x^E}{dt} = \eta[\hat{\Delta}\phi^E]_x + \mu_E\left(h(\phi_x^E) + h(\phi_x^N/\Phi_1)\right)\bar{h}(2\Omega_x) - k_E\phi_x^E,$$

$$\frac{d\phi_x^L}{dt} = \mu_L h(\phi_x^E)\bar{h}(\phi_x^N/\Phi_2) - k_L\phi_x^L,$$

$$\frac{d\phi_x^D}{dt} = \mu_D\left[h(\phi_x^E) + \bar{h}(\phi_x^N)\right]\bar{h}(2\Omega_x) - k_D\phi_x^D,$$

$$\frac{d\phi_x^N}{dt} = \left[\beta + \mu_N h([\hat{\Sigma}\phi^D]_x)\right]\bar{h}(\phi_x^D)\bar{h}(\phi_x^L) - k_N\phi_x^N. \tag{17}$$

Here, the parameters $\mu_i$ denote gain rates, $k_i$ denote decay rates, $\beta$ denotes the basal gain rate of the component N, and the $\Phi_i$ denote threshold levels for positive and negative feedbacks. The operators $\hat{\Delta}$ and $\hat{\Sigma}$ are the lattice Laplacian and the sum over concentrations of neighboring lattice sites, respectively, and defined by $[\hat{\Delta}\phi]_x = \sum_{y \in U_x}(\phi_y - \phi_x)$ and $[\hat{\Sigma}\phi]_x = \sum_{y \in U_x}\phi_y$, where $U_x$ is the set of neighbours of site $x$. The dynamics of the cell state $\Omega_x$ is given by

$$\frac{d\Omega_x}{dt} = -\frac{dV}{d\Omega}(\Omega_x) + f_{int}(\phi_x^E, \phi_x^L, \phi_x^D, \phi_x^N), \tag{18}$$

where the function $V$ is a 'potential' for the cell state which ensures that in the absence of signalling and proneural gene expression, $\Omega_x$ has two stable equilibria: $\Omega_x = 0$ (neuroepithelium) and $\Omega_x = 1$ (neuroblast) (see **Appendix 3—figure 1**). The qualitative features of our model do not depend on the exact choice of $V$. The term $f_{int}$ acts as a 'force' that triggers the transition from one state to the other depending on the local signalling activity and proneural gene expression. The functional form of $f_{int}$ is based on the observations that (i) proneural gene expression seems to be sufficient but not necessary for the transition while (ii) Notch expression seems to keep cells in their proneural state (**Yasugi et al., 2010**). Therefore, we here choose

$$V(\Omega) = \Omega^2(1-\Omega)^2/4,$$
$$f_{int}(\phi^E, \phi^L, \phi^D, \phi^N) = h(\phi^L/\Phi_{int}) + \bar{h}(\phi^N/\Phi_{int})h(\phi^E/\Phi_{int}). \tag{19}$$

The cell state potential $V$ has wells at $\Omega = 0$ (neuroepithelial state) and $\Omega = 1$ (neuroblast state) (see **Appendix 3—figure 1**), so that both states are stable and a transition occurs when the hill in between both states can be overcome. This choice of $f_{int}$ leads to a transition of the cell state from neuroepithelial ($\Omega = 0$) to neuroblast ($\Omega = 1$) when (i) L exceeds the threshold level $\Phi_{int}$ and/or (ii) E exceeds *and* N drops below the threshold level.

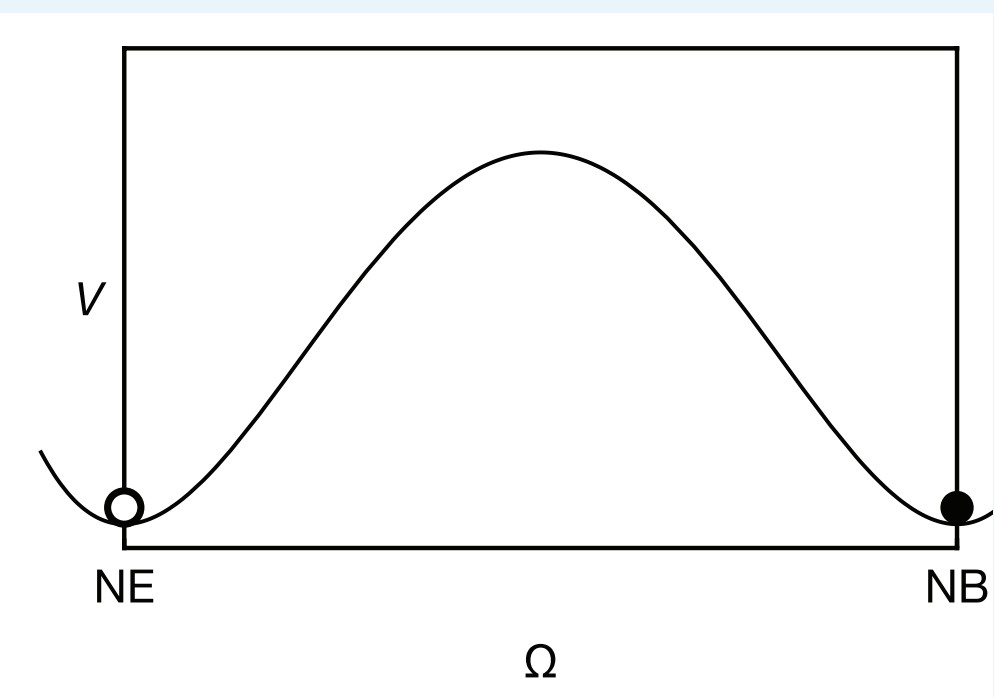

**Appendix 3—figure 1.** The cell state potential $V$, given by *Equation 19* has two wells, corresponding to the neuroepithelial state (NE, $\Omega = 0$) and the neuroblast state (NB, $\Omega = 1$).
DOI: https://doi.org/10.7554/eLife.40919.021

**Appendix 3—table 1.** Reference parameter set used for the model *Equations 17–19*.

| Parameter(s) | Value | Description | Affected components |
|---|---|---|---|
| $\eta$ | 0.02 | diffusion constant | E |
| $\mu_E$, $\mu_L$, $\beta$ | 10 | gain rates | E, L, N |
| $\mu_D$, $\mu_N$ | 5 | gain rates | D, N |
| $k_E$, $k_L$, $k_D$, $k_N$ | 1 | decay rates | E, L, D, N |
| $\Phi_1$ | 100 | Notch threshold | E |
| $\Phi_2$ | 0.5 | Notch threshold | L |
| $\Phi_{int}$ | 10 | threshold for differentiation | $\Omega$ |
| $n$ | 3 | Hill exponent | E, L, D, N |
| $\gamma$ | (as indicated) | biochemical noise strength | E, L, D, N |

DOI: https://doi.org/10.7554/eLife.40919.022

## Appendix 4

DOI: https://doi.org/10.7554/eLife.40919.015

# Robustness of the model against fluctuations and disorder

## Robustness against biochemical noise

Gene expression and biochemical reactions typically suffer from fluctuations due to small numbers of molecules involved (**Tsimring, 2014**). To achieve reliable morphological results, any biochemical mechanism governing morphogenetic processes must be robust against such types of noise. From an analytical point of view, stability of the zero signalling fixed point in our model (see **Appendix 2—figure 1**) ensures that a cell does not differentiate prematurely due to a certain degree of noise in proneural gene expression or fluctuating signalling activity. To numerically demonstrate that our model is robust against fluctuations in molecule concentrations, we performed simulations of the system in the presence of biochemical fluctuations in all four components. The noisy system is given by **Equations 17–19** with each dynamical equation replaced according to

$$\frac{\mathrm{d}\phi^i}{\mathrm{d}t} \to \frac{\mathrm{d}\phi^i}{\mathrm{d}t} + \gamma\xi_i(t)\,, \qquad (i = \mathrm{E},\mathrm{L},\mathrm{D},\mathrm{N}) \qquad (20)$$

where $\gamma$ denotes the noise strength and $\xi_i$ denotes Gaussian white noise characterised by the expectation values $\langle\xi_i(t)\rangle = 0$ and $\langle\xi_i(t)\xi_j(t')\rangle = \delta_{ij}\delta(t - t')$. Furthermore, $\delta_{ij}$ denotes the Kronecker delta and $\delta(t)$ the Dirac delta distribution.

**Appendix 4—figure 1** shows numerical examples of the system for different noise strengths $\gamma$. In these examples, the wave robustly travels from the left to the right for small and intermediate noise levels (as compared to the gain rate $\mu_\mathrm{E}$) (**Appendix 4—figure 1a,b**), whereas premature differentiation is only observed for large noise levels that introduce fluctuations comparable to physiological concentrations (**Appendix 4—figure 1c**). Finally, random differentiation throughout the tissue only occurs if the system is dominated by fluctuations (**Appendix 4—figure 1d**).

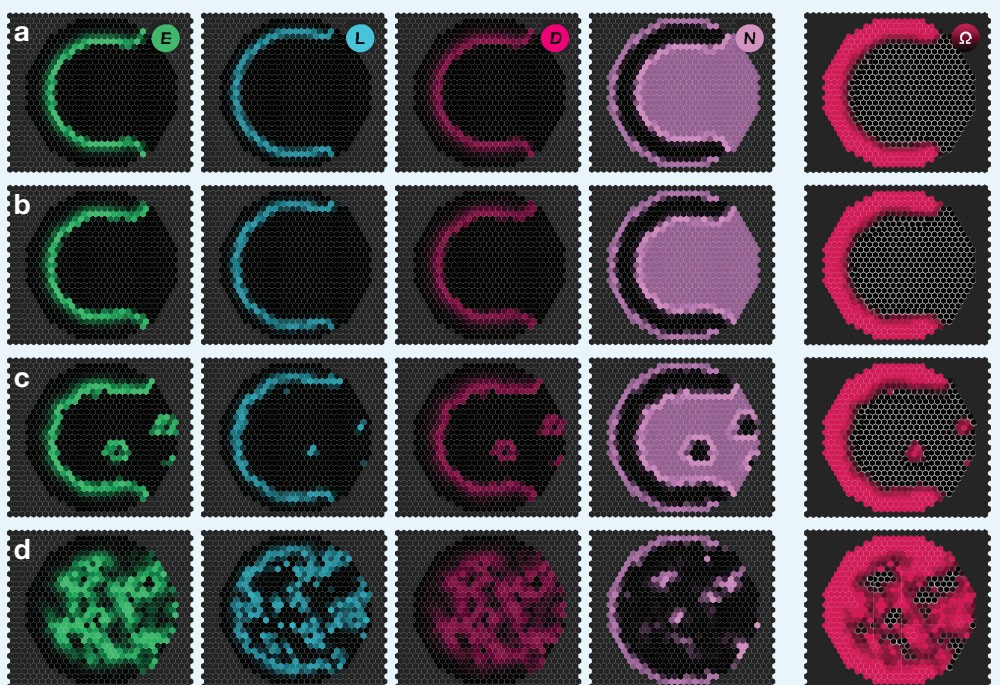

**Appendix 4—figure 1.** Response of the dynamics to biochemical fluctuations. Panels show snapshots of the system for different noise strengths: (a) $\gamma/\mu_E = 0$, (b) $\gamma/\mu_E = 0.75$, (c) $\gamma/\mu_E = 1$, (d) $\gamma/\mu_E = 1.5$. All other parameters are given in *Appendix 3—table 1*. The system given by *Equations 17–20* was simulated on a hexagonal lattice with circular geometry with a radius of 15 lattice sites. Initial conditions were localised elevated levels of E in those outer boundary cells that have angles between $\pi/3$ and $5\pi/3$ as measured from the center of the circular lattice. The respective simulation panels show snapshots of the activity of EGFR signalling (green), L'sc expression (blue), Delta activity (magenta) and Notch activity (pink), as well as the cell state $\Omega$, for which black indicates neuroepithelium ($\Omega = 0$) and red indicates neuroblasts ($\Omega = 1$). Colour intensity indicates the local gene expression levels or signalling activities, respectively. The snapshots show the time $t = 12.5$.

DOI: https://doi.org/10.7554/eLife.40919.024

## Robustness against lattice defects

To test whether the proposed mechanism is robust with respect to disordered lattice structures, we considered a site-diluted version of our model, in which randomly chosen sites in the hexagonal lattice were 'deactivated', that is removed from the dynamics and the coupling topology. Simulations showed that while the presence of disorder leads to a local distortion of the propagating front profile, the overall mechanism remains intact and leads to robust and sequential differentiation of the neuroepithelial tissue (*Appendix 4—figure 2a*). Even regions which are partially 'shielded' by defects eventually become differentiated as the diffusion-mediated propagation has no intrinsic directionality and is able to reach such regions when the wave has surrounded the corresponding region (*Appendix 4—figure 2*). Moreover, the overall phenomenology of mutant and transgenic clones remains intact on such imperfect lattices, as illustrated using the 'Notch upregulation' clone (cf. *Figure 3f* and *Appendix 4—figure 2b*).

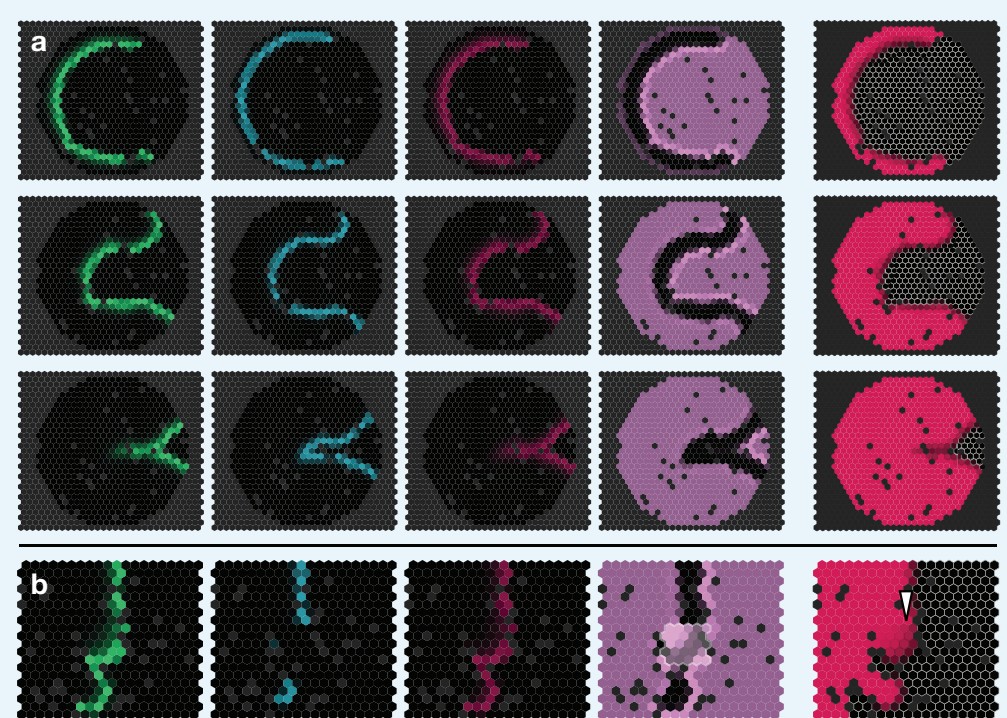

**Appendix 4—figure 2.** Simulated proneural wave on a lattice with random 'defects' (black sites). (**a**) Proneural wave propagation on a hexagonal lattice with circular geometry. Parameters and initial conditions as in *Appendix 4—figure 1* with $\gamma = 0$. The snapshots show the time $t = 10$ (top row), $t = 25$ (middle row) and $t = 40$ (bottom row). (**b**)Notch upregulation clone as shown in *Figure 3f*, but on a heavily site-diluted lattice. The white arrowhead indicates retarded differentation, cf. *Figure 3f*.

DOI: https://doi.org/10.7554/eLife.40919.025

## Appendix 5

DOI: https://doi.org/10.7554/eLife.40919.015

### Suppression of lateral inhibition patterns

An analytical argument demonstrating how basal Notch levels suppress lateral inhibition can be made in a simple picture involving Delta–Notch interactions in two cells $x = 1, 2$,

$$\frac{d\phi_x^D}{dt} = \bar{h}(\phi_x^N) - \phi_x^D \,, \qquad \frac{d\phi_x^N}{dt} = \beta + \lambda\phi_{\bar{x}}^D - \phi_x^N \,, \qquad (21)$$

where $\bar{x}$ refers to the respective other cell and $\bar{h}(\phi) \equiv 1 - h(\phi)$ with $h$ being the Hill function *Equation 5*, as before. Here, $\lambda$ is the gain rate for Notch and $\beta$ indicates the basal production. For simplicity, we consider a linear positive feedback in the dynamics of Notch and the Hill exponent $n = 2$ for $\bar{h}$. In steady state, where $d\phi_x^D/dt = 0 = d\phi_x^N/dt$, we can eliminate $\phi_1^N$ and $\phi_2^N$ to obtain

$$\phi_x^D = \Gamma(\phi_{\bar{x}}^D) \,, \qquad (22)$$

where $\Gamma(\phi) = \bar{h}(\beta + \lambda\phi)$. From this, it follows that both $\phi_1^D$ and $\phi_2^D$ satisfy $\phi_x^D = \Gamma(\Gamma(\phi_x^D))$. Among other solutions, this equation has two solutions of the form $\phi_\pm = p \pm \sqrt{q}$ with

$$p = \frac{1}{2}\frac{1}{1+\beta^2} - \frac{\beta}{\lambda} \,, \qquad q = p^2 - \frac{1+\beta^2}{\lambda^2} \,, \qquad (23)$$

which, in the case that they are real, correspond to the low-Delta/high-Notch and high-Delta/low-Notch states in adjacent cells as they satisfy $\phi_\pm = \Gamma(\phi_\mp)$.

However, bistability only exists if both solutions are real, which is the case for $q > 0$. From *Equation 23*, we find that this corresponds to

$$\lambda > 2(\beta^2 + 1)(\beta + \sqrt{\beta^2 + 1}) \,. \qquad (24)$$

*Figure 4e* shows the corresponding phase diagram for the occurrence of lateral inhibition in the two-cell system. Therefore, a basal production term, if strong enough, can prevent lateral inhibition patterns.

## Appendix 6

DOI: https://doi.org/10.7554/eLife.40919.015

### Description of mutant and transgenic clones

#### Simulation of clones

To capture experimental scenarios in which mutant or transgenic clones were induced in the neuroepithelium, we modified the model *Equation 17* accordingly. In our model, a clone is defined by a cell or a group of cells within the simulated tissue which has altered kinetic rate parameters or a different initial condition, depending on the type of experimental perturbation (*Figure 3*). For the case of downregulation of proneural gene or signalling factors, the synthesis or binding rate of the respective gene or receptor in the clone cells is decreased or set to zero, as indicated in the caption of *Figure 3*. For the case of upregulation, which in all considered cases corresponds to a constitutively active gene, we added a source term to the clone that leads to constant synthesis and furthermore set the initial condition of the clone to the elevated steady-state concentration of the respective gene or signalling activity.

#### Simulation of *Figure 7a*

To simulate the effects of clones in which EGFR signalling is constitutively activated within the neuroepithelium (*Figure 7*), we simulated the model on a hexagonal lattice with circular boundaries (*Video 4*). The initial condition was set to $\phi^{\mathrm{E}}|_{t=0} = \mu_{\mathrm{E}}/2$ in a one-dimensional array of cells in the outermost cell layer that have angles between $\pi/3$ and $5\pi/3$ as measured from the center of the circular lattice. Moreover, we arbitrarily selected four lattice sites and endowed them with a constant production of E by adding the term $\mu_{\mathrm{E}}\bar{h}(2\Omega_{\mathrm{x}})$ to the reaction dynamics of the component E in *Equation 17*; this mimics the constitutively active EGFR signalling. *Figure 7a* shows a snapshot at time $t = 14$. All parameter values are given in *Appendix 3—table 1*.

## Appendix 7

DOI: https://doi.org/10.7554/eLife.40919.015

## Sensitivity analysis of the model

### Morris method for global sensitivity analyses

To test the sensitivity of key observables on model parameters, we here employ the so-called Morris method, a widely used method for global sensitivity analyses (**Morris, 1991**; **Campolongo et al., 2007**; **Wu et al., 2013**). To briefly summarise, for a fixed model observable $\mathcal{O}$ and a given set of parameters $\theta_1, \ldots, \theta_n$, the Morris method consists in repeatedly sampling a discretised parameter space (or subspace of interest) of the model and, for each parameter $i$, calculating the so-called 'elementary effects'

$$e_i = \frac{\mathcal{O}(\theta_1, \ldots, \theta_i + \Delta, \ldots, \theta_n) - \mathcal{O}(\theta_1, \ldots, \theta_n)}{\Delta} ,\tag{25}$$

That is the finite-difference quotient of the output with respect to the parameter, given a finite step size $\Delta$ that is chosen adequately; for standard choices and further details on the method, see, for example (**Wu et al., 2013**). This sampling procedure yields a distribution $P_i(e)$ of elementary effects for each parameter $i$, from which the following sensitivity indices are computed,

$$m_i = \langle e \rangle_i , \quad m_i^* = \langle |e| \rangle_i , \quad \sigma = \sqrt{\langle e^2 \rangle_i - \langle e \rangle_i^2} ,\tag{26}$$

where $\langle \cdot \rangle_i$ denotes the expectation value under the distribution $P_i$. The interpretation of these indices is given in the main text.

### Probed observables and parameters

As output observables we here choose the linear propagation velocity $v$ of the proneural wave and the width $w$ of the transition zone. We formally define these quantities as follows for the lattice-based full proneural wave model **Equations 17–19**. By $x$, we denote the extension of the system in the direction of the travelling wave. We define $\bar{\Omega}_x = \ell_\perp^{-1} \sum_y \Omega_{xy}$ as the average of the cell state $\Omega$ in the direction perpendicular to the wave, where $\ell_\perp$ is the extension of the lattice in the perpendicular direction. A wave with constant velocity leads to a proportionally linear increase in number of neuroblasts, so that the wave velocity $v$ (in lattice sites per unit time) is given by

$$v = \frac{\mathrm{d}}{\mathrm{d}t} \sum_x \bar{\Omega}_x .\tag{27}$$

Practically, we determine $v$ as the slope obtained from a linear fit of $\sum_x \bar{\Omega}_x$ in the linear regime.

We define the width $w$ of the transition zone via the spatial spread of transitioning cells, that is those with $0 < \Omega < 1$. Formally, this width can be defined as

$$w = \left[ 2\sqrt{\pi} \sum_x \left( \bar{\Omega}_x - \bar{\Omega}_{x-1} \right)^2 \right]^{-1} .\tag{28}$$

The discrete derivative, $\bar{\Omega}_x - \bar{\Omega}_{x-1}$, which measures the steepness of the profile, is non-zero only in the transition zone. For example, for a Gaussian profile of the discrete derivative, $\bar{\Omega}_x - \bar{\Omega}_{x-1} \propto \mathrm{e}^{-x/2\sigma^2}$ with a variance $\sigma^2 \gg 1$, **Equation 28** yields $w = \sigma$. To avoid confounding effects by initial and boundary conditions in model simulations, we use the temporal median of $w$ as a proxy for the width of the transition zone.

We compute the Morris indices *Equation 26* for the dependence of $v$ and $w$ on the kinetic and diffusion properties of the integrated model, that is on the parameters $\eta$, $\beta$ and $\mu_i$, $k_i$ for $i = \mathrm{E, L, D, N}$ and allow them to vary between the 0.2-fold and 5-fold reference value given in *Appendix 3—table 1*, while keeping all other parameters fixed to their values given in *Appendix 3—table 1*. 55000 parameter samples were used to compute expectation values.

