## [Decision Letter]

Thank you for submitting your article "The proneural wave in the *Drosophila* optic lobe is driven by an excitable reaction-diffusion mechanism" for consideration by *eLife*. Your article has been reviewed by two reviewers and the evaluation has been overseen by Naama Barkai as the Reviewing and Senior Editor. The reviewers have opted to remain anonymous.

The reviewers have discussed the reviews with one another and the Reviewing Editor has drafted this decision to help you prepare a revised submission.

It is uniformly agreed that the system is interesting and the theoretical work valid. There are several issues, however, that will need to be addressed.

1) Motivating the study: given that the system was previously studied (Sato paper), it may be necessary to better define why a new model is 'needed', and what new biological insights it can bring to the table. For example, what are the mysteries that are still not understood and can be better explained by a more in-depth quantitative study? This is discussed in Appendix 3 but should be explained also in the main text.

2) Related to point 1, not all statements in Appendix 3 are supported – e.g. the statement that the Sato model is not robust. Also, the new model may not be fundamentally different from the Sato model, but rather extends it: models clearly make useful simplifications. The current work may 'open' some 'black-box' and by this provide deeper insights. This does not reduce the significance of the study. Please address that.

3) The analysis of Appendix 5 is interesting and should be part of the main text. The lack of lateral inhibition in this case presents a key biological insight that distinguishes the present study. The experimental test proposed by reviewer #1 below is therefore important.

4) Please explain all assumptions, interactions and parameters, as requested below.

Finally, please address all comments and suggestions in the detailed reviews below.

*Reviewer #1:*

This paper describes a model to explain the progression of a wave of differentiation that crosses a neuroepithelium to generate a brain structure in the optic lobes of *Drosophila*. This type of wave has been described (and modeled) quite extensively, in particular the morphogenetic furrow that patterns the *Drosophila* eye. In this case, we are dealing with the neurogenic wave that transforms epithelial cells into neural stem cells (neuroblasts) and produces one of the optic lobe structures. In fact, a recent paper had also modeled the same wave of differentiation and this paper is discussed here.

This type of patterning mechanisms is likely to be fairly common in biology and it is thus quite important to understand its fundamental aspects, which means that a comparison with the morphogenetic furrow is important to emphasize the common points and points of divergence. This will therefore be an important paper for physicists and for more theoretically oriented biologists who want to understand the rules of patterning.

This being said, as a biologist, I do not feel completely qualified to comment on the mathematics of the model. Yet, I think that the authors should have made more efforts to make more accessible the aspects of the model in which a biologist might be interested and might want to use for his/her own research. For example, it is quite difficult to find a clear description in one place of all the interactions (positive and negative) between pathway components that are actually used in the model. Figure 2D comes closest to doing so but, as with other parts of the paper, it refers to an appendix that is disconnected from the main text. The authors should therefore explain in much more precise terms the exact interactions and must justify why each arrow exists, and or why some of them (which have also been described as important for the process) have been excluded.

In particular:

- What is the evidence to support Notch promotion of EGFR signaling?

- Why isn't proneural promotion of Dl expression included?

- Is Notch repression of cell fate conversion necessary on top of the Notch inhibition of proneural factors?

- The authors do not even mention the data on JAK/STAT and Hippo affecting proneural wave progression. Why do they ignore this?

Some of these justifications have been relegated to appendices and it would make the paper much more important to the biologists if much of Appendix 3 (and similarly for Appendix 5) were to be part of the main text. The authors must do this.

There are several comparisons with the recent paper from the Sato lab, with assertions that the current model makes more accurate predictions. However, both papers use relatively crude assessments of phenotype: acceleration or retardation of the proneural wave. From the model simulations in Figure 3, the authors show that there are many 'flavors' to proneural acceleration (or retardation), i.e. different parameters can produce similar outcomes. If the authors really want to say their model is more accurate or better, then they should test model predictions appropriately (i.e. look at EGFR activity and Notch and Dl levels, in addition to Dpn and/or L'sc for the experimental conditions detailed in Figure 3. All of this should be quite easy and would add significant validation to the model.

Furthermore, the authors should also explain why their model of proneural wave progression adds to previous similar models of wave progression, in particular the morphogenetic furrow in the eye disc, also in *Drosophila*. One key feature that is distinct with the proneural wave is the lack of Notch-mediated lateral inhibition. As this was a puzzling question, the explanation that this is due to basal low levels of Notch in the neuroepithelium is very attractive. A prediction of the model (Appendix 5, Figure 1B) is that reducing Notch levels in the neuroepithelium should result in a salt-and-pepper pattern typical of lateral inhibition. This would be a key result that can very easily be tested experimentally by using a neuroepithelial driver to knock down to different extents Notch using RNAi. This would provide a qualitatively different prediction from just acceleration or retardation of the wave and would convince the biologist that the model is useful to address this point.

*Reviewer #2:*

The presented article proposes a mathematical model for a pro-neural wave in the *Drosophila* optical lobe as driven by an excitable reaction diffusion mechanism, that explains observed data and predicts a new experiment. The biochemical nature of the wave is due to a complex process, involving interaction of EGFR and Delta-Notch signaling pathways. Their interplay establishes a transition zone, that travels over the tissue and triggers a differentiation wave.

Although a phenomenological model of this process exists, the authors propose a new approach. This is justified as it helps to analyze at greater depth how this biochemical process, with interaction between multiple components, results in the wave. As such the article is an example for how mainly model driven approaches may help refining the view on a developmentally relevant, but poorly understood process.

The structure of the article is well developed. The appendix, in particular for the explanation of traveling front and pulse models, is didactically useful for a readership that has a background in life sciences.

Possible improvements:

In general, the theory is compared to data at a qualitative level. If possible, it would help to develop a more quantitative interface to experiments. For example, Figure 2E discusses a transient activation of notch signaling that is mentioned to also occur in the optic lobe. The model predicts a striking spatial profile, that may be contrasted to a similar profile of data for a few of the genes involved.

Further, Figure 3 shows a comparison of model simulations on a 2D lattice to experiment. One of the striking differences is that the simulation is based on an ordered hexagonal lattice, while data looks like a strongly disordered array. Order may be less important in case of strong effects such as shown in panels A,B,D. It remains unclear however, how the underlying lattice order will affect the more subtle effects shown in the cell state variable for panels C,E,F.

In Appendix 3, the authors explain the concrete realization that *H_0_* is not important to capture the qualitative features of the model. However, in the spirit of the well-developed didactic tone in the preceding appendices, the model might become more transparent to a broader audience, if key features of *H_0_* are further developed. The mathematics is clear, but additional graphics and a brief discussion would help to guide the intuition of this aspect.

The following Appendix 4 presents a valuable in-depth analysis about robustness against fluctuations. However, it is difficult to find a more detailed discussion about choice of parameters presented in Appendix 3—Table 1. Are these parameters chosen in a physiologically relevant regime? What's the effect of changes to the model? For future experimentalists aiming to test this model, it may be good to provide a range of possible parameter values in which the presented features will be valid. For example, is there a microscopic motivation for the choice of the Hill coefficient n = 3?

This is an exciting model, and it would be very helpful for future development if the model could provide additional predictions that are potentially difficult to test using current technology. For example, could the model be used to predict what sets the speed of the wave? This point is not essential to support the major conclusions of the article. But in the interest of theory motivating new experiments, such an addition might be helpful.

---

## [Author Response]

It is uniformly agreed that the system is interesting and the theoretical work valid. There are several issues, however, that will need to be addressed,1) Motivating the study: given that the system was previously studied (Sato paper), it may be necessary to better define why a new model is 'needed', and what new biological insights it can bring to the table. For example, what are the mysteries that are still not understood and can be better explained by a more in-depth quantitative study? This is discussed in Appendix 3 but should be explained also in the main text.

We have expanded the Introduction and Discussion section of our manuscript as described below.

First, the paper by Sato et al., has clearly pioneered a modelling approach on proneural wave progression, however, it remains vague on the molecular basis of the wave. It states that EGFR signalling triggers differentiation, while differentiation triggers EGFR signalling activity, resulting in essentially a two-state system that propagates by construction. It does not address the emergence of major characteristics of the wave such as spatially confined proneural gene expression in a localised transition zone. Moreover, a concrete molecular mechanism and/or cascade that could be pinpointed (and therefore falsified) or reenacted, does not exist. To highlight this more clearly, we have expanded the corresponding section in the Introduction, mentioning the lack of understanding of the following four major points:

i) the dynamical nature of the wave,

ii) the emergence of a localised transition zone with spatially confined expression of the proneural gene, *l’sc*

iii) the specific profiles of gene expression and signalling activity around the transition zone,

iv) the effects of molecular kinetics on wave propagation,

In the Discussion section, we have added new sections to highlight the specific explanatory and mechanistic novelties.

In addition, the Sato model exhibits some rather puzzling aspects in its dynamical implementation, such as the linearity of Delta-Notch interactions and its lack of resilience to noise. Rather than provide a critical discussion of these more detailed aspects in the main text, this comparison is discussed in Appendix 3.

2) Related to point 1, not all statements in Appendix 3 are supported – e.g. the statement that the Sato model is not robust.

This point refers to the following sentence in our manuscript: "Moreover, the resulting model is unstable with respect to additive fluctuations in gene expression and signalling activity; slight misexpression or perturbations in signalling activity will result in an immediate premature differentiation."

Indeed, this statement is already made, in a more formal way, in the original paper by Sato et al., 2016: “Because the equation for *A* is unstable at *A* = 0, the simple addition of noise induces spontaneous differentiation apart from the wave front.”

(Here, *A* is the variable encoding both the cell state and proneural gene expression; in the words of the authors: “*A* is an abstract value specifying the state of differentiation […]. We assume that *A* is intimately related to the expression levels of AS-C proteins.” The instability arises due to the fact that any perturbation of A away from 0 drives A towards the differentiated state if EGF expression is sufficiently large.)

Since a repetition of the formal argument already given in the original reference would be redundant, we have added "as discussed in Sato et al., (2016)." to the corresponding sentence in Appendix 3.

Also, the new model may not be fundamentally different from the Sato model, but rather extends it: models clearly make useful simplifications. The current work may 'open' some 'black-box' and by this provide deeper insights. This does not reduce the significance of the study. Please address that.

Our model is formulated in a framework of reaction-diffusion systems similar to that used in Sato et al. and we do not dispute that this is a sensible and natural framework to describe the process at hand. However, we do not believe that our model is an extension or unfolding of the Sato model. The Sato model is, in vital parts, based on the phenomenology of the wave. To exemplify this, let us look at the formal mechanism that leads to stabilisation of the wave in Sato et al., (2016). EGF is diffusible but the wave can only propagate if EGF activity is directly or indirectly activated:

“However, EGF ligand and Dl are only produced at the interface between undifferentiated and differentiated cells but not produced in differentiated NBs. Thus, we include the EGF and Dl production terms *a_e_ A(A_0_ – A)* and *a_d_ A(A_0_ – A)*, respectively.” (Sato et al., 2016).

It is not straightforward to imagine the biochemical basis of such an interaction, which activates EGF signalling for intermediate values of the abstract differentiation variable *A*. Importantly, our model neither builds upon nor expands on such or similar notions.

Instead, our model is constructed by gradually building up the phenomenology of the wave from known molecular interactions. To our knowledge, the idea that autocrine EGF signalling drives the proneural wavefront and that differentiation shuts down this driver mechanism and renders it excitable has not been presented before. It is certainly not implicit in the model of Sato et al., where in fact the molecular mechanism of wave propagation remains rather cryptic for the above reasons. We would therefore argue that our model provides conceptually different insights rather than a refinement of previous work.

3) The analysis of Appendix 5 is interesting and should be part of the main text. The lack of lateral inhibition in this case presents a key biological insight that distinguishes the present study. The experimental test proposed by reviewer #1 below is therefore important.

We have moved the corresponding sections about the suppression of lateral inhibition patterns and the role of cis-inhibition to the main text, together with the corresponding simulation figures, which constitute the new Figure 4 in the main text.

4) Please explain all assumptions, interactions and parameters, as requested below.

In the revised manuscript, we have partitioned the original section on the full model of the proneural wave into two separate sections, one of which explains the structure of the full model (including a reference for each interaction) shown in Figure 2D.

Finally, please address all comments and suggestions in the detailed review below.Reviewer #1:[…] This will therefore be an important paper for physicists and for more theoretically oriented biologists who want to understand the rules of patterning.This being said, as a biologist, I do not feel completely qualified to comment on the mathematics of the model. Yet, I think that the authors should have made more efforts to make more accessible the aspects of the model in which a biologist might be interested and might want to use for his/her own research. For example, it is quite difficult to find a clear description in one place of all the interactions (positive and negative) between pathway components that are actually used in the model. Figure 2D comes closest to doing so but, as with other parts of the paper, it refers to an appendix that is disconnected from the main text. The authors should therefore explain in much more precise terms the exact interactions and must justify why each arrow exists, and or why some of them (which have also been described as important for the process) have been excluded.

Please see our response to point 4 from the reviewing editor.

In particular:- What is the evidence to support Notch promotion of EGFR signaling?

It has been shown previously that expression of N^act^ results in the activation of PntP1 (Yasugi et al., 2010, see Figure 6D,D').

- Why isn't proneural promotion of Dl expression included?

As a starting point for representing Delta-Notch signalling in our model, we had originally chosen the well-established model of (Collier et al., 1996), which includes the effective feedback between Notch and Delta (via proneural genes) as a direct interaction between effective variables. In fact, we have checked in simulations that an additional explicit promotion of Dl expression through proneural genes does not alter the phenomenology of the model. This is due to the fact that such an additional interaction would only reinforce an already existing feedback: Notch would suppress *l'sc*, which, in turn, would lead to decreased activation of Delta, i.e., would provide a negative feedback that is already present. A high degree of functional redundancy with respect to additional interactions is common in fairly complex genetic regulatory networks, which generically fall into the “sloppiness universality class” (see, e.g., the excellent papers by Gutenkunst et al., 2007; Daniels et al., 2008; Waterfall et al., 2009). To maintain the clarity of what is already a complex model, we have therefore opted to avoid such manifest redundancies.

- Is Notch repression of cell fate conversion necessary on top of the Notch inhibition of proneural factors?

It is known from the literature that expression of the proneural gene *l'sc* is sufficient but not necessary for the NE-to-NB transition (Yasugi et al., 2008; Egger et al., 2010; Sato et al., 2013). Furthermore, Notch is thought to regulate the width of the transition zone, making the spatially confined region within the transition zone (in which Notch activity is downregulated) a prime candidate for a region "permissive" for transition. In this spirit, Notch acts as an inhibitor of the transition in our model. Otherwise, EGFR signalling alone would be able to initiate the transition, for which we have no reason to assume is the case.

- The authors do not even mention the data on JAK/STAT and Hippo affecting proneural wave progression. Why do they ignore this?

While we are aware that the JAK/STAT and Hippo pathways modulate proneural wave progression, neither seems to be a crucial driver of the wave itself, nor do their expression patterns suggest a concomitant motion with the wave. Rather, the broad expression of their ligands and/or receptors in the neuroepithelium suggests that they act as modulators of proneural wave progression that prevent premature transition (in the case of JAK/STAT) and to regulate the mitotic activity of neuroepithelial cells (in the case of Hippo) (Yasugi et al., 2008; Reddy et al., 2010; Sato et al., 2013). This is why we refrained from including them in our manuscript. In the revised manuscript, we have added to the Discussion section a paragraph on these pathways, elaborating on their role in modulating proneural wave progression, while not being drivers of the wave, as suggested by experiments.

Some of these justifications have been relegated to appendices and it would make the paper much more important to the biologists if much of Appendix 3 (and similarly for Appendix 5) were to be part of the main text. The authors must do this.

As explained above, we have partitioned the "full model" section in the manuscript into two sections, the first of which ("Integrated model of the proneural wave") is devoted to explaining all model interactions including references. The second section ("Congruence with experimental data") contains the considerations about the suppression of lateral inhibition patterns that were originally part of Appendix 5, as suggested by the referee. We believe that this increases the clarity of the manuscript and highlights the significance of the findings. Nevertheless, we have opted to keep the formal, more mathematical aspects of the full model formulation and the suppression of lateral inhibition in the respective appendices, where they are still available to the interested reader.

There are several comparisons with the recent paper from the Sato lab, with assertions that the current model makes more accurate predictions. However, both papers use relatively crude assessments of phenotype: acceleration or retardation of the proneural wave. From the model simulations in Figure 3, the authors show that there are many 'flavors' to proneural acceleration (or retardation), i.e. different parameters can produce similar outcomes. If the authors really want to say their model is more accurate or better, then they should test model predictions appropriately (i.e. look at EGFR activity and Notch and Dl levels, in addition to Dpn and/or L'sc for the experimental conditions detailed in Figure 3. All of this should be quite easy and would add significant validation to the model.

We have performed a general sensitivity analysis of the model, which illuminates the influence of a parameter on key observables of the system such as the proneural wave speed and the width of the transition zone. Many of these parameters refer to interactions that are not present in the Sato model, such as the ones entailed by independent variables for proneural gene expression and differentiation status.

Furthermore, the authors should also explain why their model of proneural wave progression adds to previous similar models of wave progression, in particular the morphogenetic furrow in the eye disc, also in Drosophila.

In the revised manuscript, we have expanded the paragraph in the Discussion mentioning progression of the morphogenetic furrow as a similar process, but also pointing out that the eye disc exhibits quite different features with regard to growth and deformation of the underlying tissue as well as the network structure that leads to propagation of the furrow.

One key feature that is distinct with the proneural wave is the lack of Notch-mediated lateral inhibition. As this was a puzzling question, the explanation that this is due to basal low levels of Notch in the neuroepithelium is very attractive. A prediction of the model (Appendix 5, Figure 1B) is that reducing Notch levels in the neuroepithelium should result in a salt-and-pepper pattern typical of lateral inhibition. This would be a key result that can very easily be tested experimentally by using a neuroepithelial driver to knock down to different extents Notch using RNAi. This would provide a qualitatively different prediction from just acceleration or retardation of the wave and would convince the biologist that the model is useful to address this point.

We tested this prediction by lowering Notch levels in the neuroepithelium but did not observe 'salt-and-pepper' patterns of Delta/Notch expression within clones expressing Notch RNAi (see the new Figure 4C and D). However, the absence of the emergence of lateral inhibition is likely due to the complete loss of detectable Notch in cells expressing Notch RNAi, while the reduction of Notch levels in the model prediction is more subtle. Referring to the 'phase diagram' in the new Figure 4E, it can be seen that both basal and Delta-regulated Notch activity need to be in the appropriate range for lateral inhibition patterns to occur, which is difficult to achieve experimentally. Furthermore, our model entails that Notch downregulation is a necessary (but not generally sufficient) condition for inducing salt-and-pepper patterns.

Reviewer #2:[…] The structure of the article is well developed. The appendix, in particular for the explanation of traveling front and pulse models, is didactically useful for a readership that has a background in life sciences.Possible improvements:In general, the theory is compared to data at a qualitative level. If possible, it would help to develop a more quantitative interface to experiments. For example, Figure 2E discusses a transient activation of notch signaling that is mentioned to also occur in the optic lobe. The model predicts a striking spatial profile, that may be contrasted to a similar profile of data for a few of the genes involved.

We have included a new image to show this more clearly. Please see Figure 1C.

Further, Figure3 shows a comparison of model simulations on a 2D lattice to experiment. One of the striking differences is that the simulation is based on an ordered hexagonal lattice, while data looks like a strongly disordered array. Order may be less important in case of strong effects such as shown in panels A,B,D. It remains unclear however, how the underlying lattice order will affect the more subtle effects shown in the cell state variable for panels C,E,F.

While the reviewer is right that, in general, the underlying lattice order can have strong effects on patterning phenomena, especially in systems with repulsive interactions where defects and frustration effects can play an important role (e.g., in those cases where lateral inhibition does occur!), the lattice structure is less important in diffusive systems with suppressed repulsion such as the one reported here. Hence, we have no reason to assume that the effects of clones illustrated here would be systematically different on different lattice structures. If this were the case, even the results of clonal experiments would depend on the specific local topology of the clone for which we have no indication. To illustrate this point, we have performed simulations on site-diluted lattices that introduce random defects in the lattice structure. These simulations have been included in the robustness analysis in Appendix 4, showing that both the general mechanism of wave propagation as well as the phenomenology of the more subtle clonal effects remain intact, as illustrated using the clonal scenario in which Notch has been upregulated.

In Appendix 3, the authors explain the concrete realization that H_0_ is not important to capture the qualitative features of the model. However, in the spirit of the well-developed didactic tone in the preceding appendices, the model might become more transparent to a broader audience, if key features of H_0_ are further developed. The mathematics is clear, but additional graphics and a brief discussion would help to guide the intuition of this aspect.

To improve the motivation for the choice of this function that encapsulates the two possible cell states (neuroepithelial and neuroblast), we have replaced the function *H*_0_ by a 'potential' function whose derivative is *H*_0_. We have added an additional figure in Appendix 3, where it is referred to, explaining the interpretation of this potential function in the spirit of the preceding appendices, cf. Appendix 1—Figure 1B. We believe this has improved the clarity of the model description.

The following Appendix 4 presents a valuable in-depth analysis about robustness against fluctuations. However, it is difficult to find a more detailed discussion about choice of parameters presented in Appendix 3—Table 1. Are these parameters chosen in a physiologically relevant regime? What's the effect of changes to the model? For future experimentalists aiming to test this model, it may be good to provide a range of possible parameter values in which the presented features will be valid. For example, is there a microscopic motivation for the choice of the Hill coefficient n = 3?This is an exciting model, and it would be very helpful for future development if the model could provide additional predictions that are potentially difficult to test using current technology. For example, could the model be used to predict what sets the speed of the wave? This point is not essential to support the major conclusions of the article. But in the interest of theory motivating new experiments, such an addition might be helpful.

First, we would like to mention that the core phenomenology of the model is very robust and does not depend on the precise choice of the model parameters unless some clear separation of time scales (e.g., in the kinetic parameters of the different components) is introduced, which can prevent proneural wave progression. This also holds for the Hill coefficient, which sets the steepness of the feedback.

Having said that, we found it interesting to probe the influence of the model parameters in a systematic way, especially with respect to key observables such as the speed of the wave (as mentioned by the reviewer) but also the width of the transition zone. To this end, we have performed a comprehensive sensitivity analysis of the model using the so-called 'Morris method' (Morris, 1991). This method evaluates the impact of the kinetic rate parameters and diffusion constants of the different model components on the output quantities of interest. The results of this new analysis are presented in the main text and Figure 5 of the revised manuscript with technical details deferred to the new Appendix 7. We believe that this additional analysis strengthens the predictive value of the model and can guide the direction of further experiments. We thank the reviewer for this suggestion.